# Self-Supervised Motion Magnification by Backpropagating Through Optical Flow

**Zhaoying Pan**[*]    **Daniel Geng**[*]    **Andrew Owens**
University of Michigan
https://dangeng.github.io/FlowMag

## Abstract

This paper presents a simple, self-supervised method for magnifying subtle motions in video: given an input video and a magnification factor, we manipulate the video such that its new optical flow is scaled by the desired amount. To train our model, we propose a loss function that estimates the optical flow of the generated video and penalizes how far if deviates from the given magnification factor. Thus, training involves differentiating through a pretrained optical flow network. Since our model is self-supervised, we can further improve its performance through test-time adaptation, by finetuning it on the input video. It can also be easily extended to magnify the motions of only user-selected objects. Our approach avoids the need for synthetic magnification datasets that have been used to train prior learning-based approaches. Instead, it leverages the existing capabilities of off-the-shelf motion estimators. We demonstrate the effectiveness of our method through evaluations of both visual quality and quantitative metrics on a range of real-world and synthetic videos, and we show our method works for both supervised and unsupervised optical flow methods.

## 1   Introduction

Motion magnification methods [32, 58, 40] increase the size of tiny motions in a video, revealing subtle details that are difficult to discern with the naked eye. However, existing methods come with significant limitations. Early hand-crafted methods generally require small periodic motions [64, 58] or a human in the loop [32]. More recent supervised learning methods [40] require ground-truth training examples, such as videos before and after magnification, which are difficult to obtain without synthetic data. Creating this synthetic data is a challenging problem, since it seemingly requires capturing all of the possible objects and motions that one might ever want to magnify.

In parallel, the field of motion estimation has addressed many closely related challenges. Modern optical flow networks [65, 54, 51, 10] are designed to track objects undergoing complex motions, both large and small. Most of these methods are trained on supervised datasets that capture a wide variety of objects, but parallel work has shown the effectiveness of unsupervised flow estimation [33, 27]. We ask whether we can use motion estimation models to train *magnification* models, taking advantage of their existing capabilities and reducing the need for special-purpose training data.

We propose a simple motion magnification model whose supervision signal comes from an off-the-shelf motion estimation model. Our method exploits the fact that optical flow networks are differentiable, and thus can be used as part of a loss function. We train a model to take a pair of video frames and a magnification factor $\alpha$ as input, and to generate a new pair whose predicted optical flow is $\alpha$ times as large as that of the input. We simultaneously optimize a regularization loss that preserves the visual appearance of each tracked pixel in the generated video. Notably, our model can

---

[*]Equal contribution.

37th Conference on Neural Information Processing Systems (NeurIPS 2023).

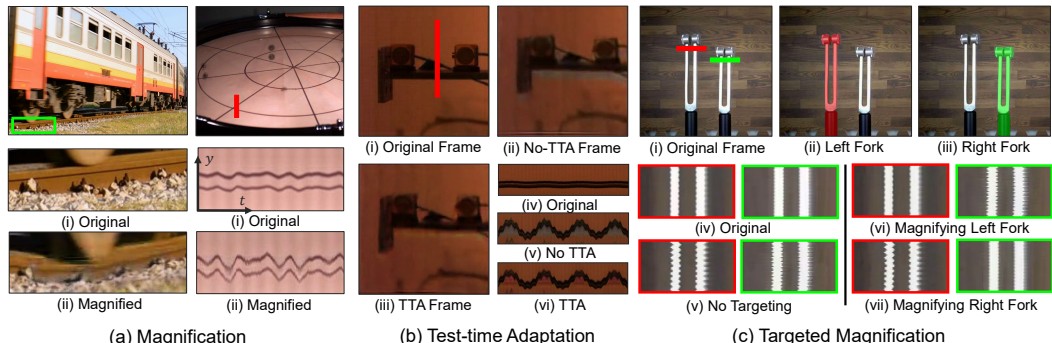

Figure 1: **Magnifying Motions in Video.** **(a)** We show frames along with closeups (Left) or $y$-$t$ slices (Right) from original videos and magnified videos. The spatial location of the $y$-$t$ slice is visualized by superimposing a colored line on to the still frames, and the location of closeups is shown with a rectangle colored green. **(b)** We show the frames from the original video, the magnified frame (No-TTA), the magnified frame after test-time adaptation (TTA), and the $y$-$t$ slices of the three videos. Test-time adaptation (i.e., finetuning on the input video) improves generation quality as can be seen in the $y$-$t$ slices. **(c)** We magnify the motions within a user-selected object segmentation [29] and show $y$-$t$ slices of the magnified videos on different targets. We use red/green colored lines to indicate the locations of $y$-$t$ slices for different targets.

be trained solely using real unlabeled videos. The optical flow estimator is only used within the loss function during training—we do not require it at test time.

Our model's simplicity and its ability to be trained solely through self-supervision provides it with several advantages over other approaches. We show that we can improve our generated videos through test-time adaptation [52, 26] by finetuning on a given input video. Our formulation can also be easily extended to magnify the motions of a single object, specified via a user-provided mask. In addition, we show results using both off-the-shelf supervised [54] and unsupervised [33] optical flow methods.

Our method is closely related to Lagrangian magnification methods [32, 19], which tracks individual pixels through a video and then amplifies their motion. While this approach is intuitively appealing and was the basis for the earliest magnification methods, a major shortcoming is that it is not clear how to combine the tracking and synthesis together. Previous Lagrangian approaches [32] often rely on hand-crafted pipelines that combine motion estimation, warping, and inpainting steps. By contrast, our method avoids these pitfalls by using off-the-shelf image-to-image translation architectures [45], and uses tracking only within the loss function.

We demonstrate the effectiveness and flexibility of our model in several ways. First, we show through experiments on real and synthetic videos that the optical flow in the generated videos more closely matches the desired motion than videos generated by baseline approaches, such as methods based on warping or supervised learning [40]. Second, we show qualitatively that our model can successfully magnify motions for a variety of videos, containing both small and large motions. Finally, we show that we can improve generation quality using test-time adaptation, and magnify individual objects using user-supplied segmentation maps.

## 2   Related Work

**Lagrangian magnification.**    Motion magnification methods can be broadly categorized as either Lagrangian [32] or Eulerian [64], a classification borrowed from fluid dynamics. Lagrangian methods explicitly track pixels, then generate new frames by forward warping (i.e., splatting) using magnified velocity estimates. Liu *et al.* [32] originally proposed the motion magnification problem and solved it using a Lagrangian approach. They modeled motion as feature trajectories and (with human-in-the-loop assistance) cluster pixels by similarity in position, color, and motion. Recent methods have extended Lagrangian magnification with more accurate optical flow [17]. While our approach performs explicit tracking and thus is closely related to Lagrangian methods, we decouple generation from tracking: we produce images using off-the-shelf networks and track *only within the loss function*. This allows us to avoid some of the challenges of the "warp and inpaint" approach, such as handling occlusions and filling holes.

**Eulerian magnification.**    Eulerian methods magnify motions without explicit motion estimation. Instead, they generate a magnified video by amplifing the temporal changes at fixed locations/pixels. Wu *et al.* [64] decomposed a video into frequency bands and applied temporal filters to extract a signal at a specific bandpass. The extracted signal is then amplified by a magnification factor and added back to the video. Wadhwa *et al.* [58] proposed a phase-based method, using complex steerable pyramids [47] to decompose the video and separate the phase from the amplitude to amplify the temporally-bandpassed phases. Later work improves efficiency using Riesz pyramids [59] and removes large motion by decomposing a scene into foreground and background layers [11]. Zhang *et al.* [72] magnified small motions while ignoring large motions by amplifying the motion field acceleration using second-order temporal filters. Concurrent work [15] extended 2D Eulerian methods to 3D motion magnification. Eulerian methods are well-suited to tracking small motions at high spatial frequencies, and may struggle when handling large motions [58], whereas Lagrangian methods' magnification quality is determined by the quality of the optical flow predictions [64]. Since our method's loss function is defined using optical flow, its capabilities are more similar to those of Lagrangian methods.

**Learning-based magnification.**    Several recent works have proposed learning-based motion magnification models. Oh *et al.* [40] created a synthetic dataset containing image segments extracted from PASCAL VOC [12] as moving foreground and images from COCO [30] as background, and trained a model to regress a ground truth magnified image from two video frames. Inspired by the steerable pyramid used in Eulerian methods [58], they also proposed an architecture that has inductive biases that encourage it to generate crisp images akin to those in Eulerian magnification, by decomposing the image into a shape and texture representation. Other work has extended this supervised magnification approach using 3D CNNs [28] and lightweight architectures [48], and has magnified microexpressions using attention-based models [63]. In contrast to these approaches, our method is *self-supervised* and learns from unlabeled videos. Recently, Gao *et al.* [19] use a very similar supervised learning objective and dataset but augment the model of Oh *et al.* [40] with inductive biases that encourage Lagrangian-like magnification by adding an attention map that is guided by an optical flow field. Instead of making optical flow part of our architecture, as an inductive bias that aids supervised training, we use it to define a self-supervised *loss function*.

**Motion estimation.**    The field of motion estimation is closely related to motion magnification. Early work solved linearized models after making color constancy assumptions [35] or solved smoothness-regularized models with variational methods [23]. Later work added robust losses and inference strategies [49, 3, 31, 14]. While these methods are well-suited to the subtle motions considered in motion magnification, it is difficult to use them as part of a loss function (as we do in this work), since it is not straightforward to differentiate through during gradient-based learning. Recent methods based on neural networks address this issue, since they are differentiable and highly accurate. A variety of recent methods train neural networks using real or synthetic optical flow data [16, 25, 43, 24, 67, 51, 65] or unlabeled video data [71, 44, 61, 34, 27, 2]. Our use of a pretrained flow model removes the need for special-purpose supervised motion magnification datasets.

**Optical flow within a loss function.**    We take inspiration from recent work that uses differentiable optical flow models to define loss functions that internally perform motion reasoning. Geng *et al.* [20] used flow to obtain robustness to small positional errors on image generation tasks. Goyal *et al.* [21] used flow to measure the distance to a goal state for robotic planning. Other work has backpropagated through optical flow as part of a pose estimation model [55]. By contrast, we use flow to compare the motions in two videos.

## 3   Method

We describe traditional Lagrangian motion magnification, and our proposed self-supervised Lagrangian magnification model.

### 3.1   Lagrangian Motion Magnification Overview

Lagrangian motion magnification methods magnify motion by tracking pixels over time, then resynthesizing the video such that the motion of each pixel has increased by a desired amount. More concretely, a point $\mathbf{x}$ in the initial frame $I_0$ of a video might be displaced by motion field $\mathcal{T}(\mathbf{x}; t)$ in

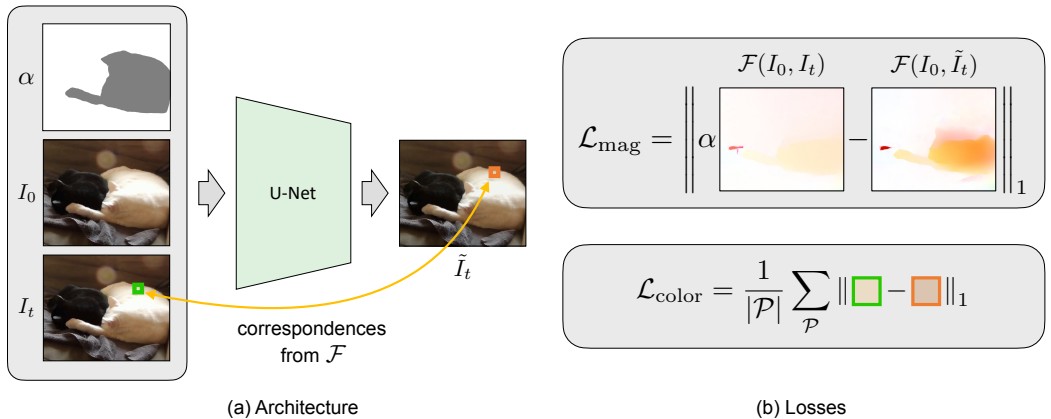

| (a) Architecture | (b) Losses |

Figure 2: **Motion Magnification Model.** Given a reference frame $I_0$, a frame to be magnified $I_t$, and a map of per-pixel magnification factors $\alpha$, we predict a magnified frame $\tilde{I}_t$. We minimize two losses that each use an off-the-shelf optical flow estimator $\mathcal{F}(\cdot, \cdot)$. First, we use a magnification loss $\mathcal{L}_{\text{mag}}$ that encourages the optical flow of the generated video to be $\alpha$ times as large as that of the input video. And second, we also include a consistency loss $\mathcal{L}_{\text{color}}$ that measures the visual similarity of corresponding pixels in $I_t$ and $\tilde{I}_t$.

frame $t$. Assuming color constancy, we have:

$$I_t(\mathbf{x} + \mathcal{T}(\mathbf{x}; t)) = I_0(\mathbf{x}), \tag{1}$$

where $I_t$ is the frame at time $t$ and $I(\cdot)$ indicates pixel access. To generate a version of the video magnified by a factor $\alpha$[1], one could synthesize a new frame $\tilde{I}_t$ by increasing the distance traversed by each pixel:

$$\tilde{I}_t(\mathbf{x} + \alpha \mathcal{T}(\mathbf{x}; t)) \leftarrow I_0(\mathbf{x}). \tag{2}$$

For example, early work in Lagrangian motion magnification [32] aimed to achieve this by performing forward warping (splatting). However, this leads to holes and aliasing artifacts [53]. Moreover, there is no mechanism for dealing with occlusions: after warping, background pixels may move in front of the foreground. Finally, such a model would not be able to deal with appearance changes without additional constraints (e.g., due to lighting variation).

## 3.2  Self-Supervised Lagrangian Magnification

We propose a motion magnification model that avoids the limitations of previous Lagrangian methods, and that can be trained solely using unlabeled video (Figure 2). Consider $\mathcal{F}(I_0, I_t)$, the motion field obtained by computing optical flow between a reference frame $I_0$ and another frame $I_t$. Our goal is to generate a magnified frame $\tilde{I}_t$ such that its predicted flow is $\alpha \mathcal{F}(I_0, I_t)$, where $\alpha$ is the magnification factor. To do this, we optimize an objective to minimize the difference between the estimated flow $\mathcal{F}(I_0, \tilde{I}_t)$ and the desired flow:

$$\mathcal{L}_{\text{mag}} = \|\alpha \mathcal{F}(I_0, I_t) - \mathcal{F}(I_0, \tilde{I}_t)\|_1, \tag{3}$$

which we term the *magnification loss*. When $\mathcal{F}$ is implemented as a neural network, we can optimize this loss using gradient-based learning methods.

Optical flow models are invariant to a variety of photometric changes (e.g., changes in illumination). Thus, optimizing Eq. 3 alone would result in an underconstrained problem. To ensure that the generated frames match the colors in the input video and to also regularize our problem, we use an additional *color loss*. This loss ensures that corresponding pixels in generated frame $\tilde{I}_t$ and original frame $I_t$ are the same color. To enforce this, we put both frame $I_t$ and $\tilde{I}_t$ into correspondence with the common reference frame $I_0$ by backward warping each frame with their respective flows: $\mathcal{F}(I_0, I_t)$ and $\mathcal{F}(I_0, \tilde{I}_t)$. We then measure the distance between the warped frames. This results in a loss:

$$\mathcal{L}_{\text{color}} = \|\text{warp}(I_t, \mathcal{F}(I_0, I_t)) - \text{warp}(\tilde{I}_t, \mathcal{F}(I_0, \tilde{I}_t))\|_1. \tag{4}$$

---

[1] Note that some previous works [64, 58, 40] define the magnification factor in a way that is equivalent to $\alpha + 1$ (instead of $\alpha$), which is more conducive to their theoretical analysis.

This loss also disincentivizes adversarial examples against the flow network. Our full loss is a weighted sum of these two losses:

$$\mathcal{L} = \mathcal{L}_{\mathrm{mag}} + \lambda_{\mathrm{color}}\mathcal{L}_{\mathrm{color}}. \tag{5}$$

One benefit of putting optical flow into the loss function is that our model needs access to the flow network only during training; at inference time the model simply takes in two frames and outputs a magnified frame. Pseudocode for our method for training and inference can be found in Algorithm 1 and Algorithm 2 respectively.

### 3.3 Image Generation Architecture

One benefit of our formulation is that our proposed loss (Eq. 5) is independent of the image generation architecture, so that "off-the-shelf" image translation architectures can be used to create a magnified image. This is in contrast to other learning-based methods, which incorporate inductive biases within the network to generate images that closely resemble the input (e.g., by altering the structure of an image while preserving the texture [40]). To generate magnified frames, we use a standard U-Net architecture [45] that takes two input frames (the reference frame and the frame to be magnified) concatenated channel-wise. We encode magnification factor $\alpha$ using a sinusoidal positional embedding [39] and tile it to match the same spatial dimensions as the input frames. We concatenate this embedding channel-wise to the input frames. Please see Section A2 in the appendix for full architectural details.

### 3.4 Targeted Magnification

Our model gives us the ability to vary the magnification factor spatially within an image. We achieve this by providing different values of $\alpha$ for each pixel at inference time. This works even when training with spatially constant alpha maps, due to the fully convolutional nature of our network[2]. As a special case, we magnify the motion of a single object by setting the magnification factor to a value of $\alpha$ within an object segment, and to 1 everywhere else. In practice, we use the recent Segment Anything Model (SAM) [29] to extract a mask for a given object in the reference frame and then dilate the segmentation mask by a small fixed number of pixels. We provide pseudocode for targeted magnification during inference in Algorithm 2.

**Algorithm 1** Pseudocode in a PyTorch-like style for training a U-Net for motion magnification.

```
# Load data of two-frame videos
for (im0, im1) in dataloader:
    # Sample alpha and get embedding
    a = sample_alpha(min_alpha, max_alpha)
    pe_a = positional_embedding(a)
    pe_a = spatial_tile(pe_a)

    # Predict magnified frame with a network
    input = concat([im0, im1, pe_a])
    im1_mag = UNet(input)

    # Estimate the motion
    F_src = optical_flow(im0, im1)
    F_tgt = optical_flow(im0, im1_mag)

    # Warp the second frame
    warp_im1 = warp(im1, F_src)
    warp_im1_mag = warp(im1_mag, F_tgt)

    # Calculate losses
    mag_loss = l1_loss(a * F_src, F_tgt)
    color_loss = l1_loss(warp_im1, warp_im1_mag)
    loss = mag_loss + weight * color_loss

    loss.backward()
    optimizer.step()
```

**Algorithm 2** Pseudocode in a PyTorch-like style for inference (targeted magnification).

```
im0 = video[0]
magnified_video = [im0]
# Load frames from the input video
for frame in video[1:]:
    # Get masked embedding for input alpha
    pe_a = positional_embedding(a)
    pe_a = spatial_tile(pe_a)
    pe_a = mask * pe_a

    # Predict magnified frame with a network
    input = concat([im0, frame, pe_a])
    mag_frame = UNet(input)
    magnified_video.append(mag_frame)
```

### 3.5 Test-Time Adaptation

Since our model is entirely self-supervised, we can finetune it at inference time on the input video [52, 26, 18]. This allows us to adapt to new motions and content. We find that this can significantly improve results, especially for out-of-domain videos. Previous supervised approaches do not have this capability as they require labeled data. To perform test-time adaptation we take the frames in a given inference video, apply minor cropping and color augmentations, and finetune the model with our loss.

---

[2] While the model could be trained with random spatially-varying $\alpha$ values, we did not see a qualitative improvement when doing this.

# 4 Experiments

We evaluate our model both quantitatively through experiments with real and synthetic data, and qualitatively on real videos. We highly encourage the reader to view our website, since the magnified motions can be challenging to visualize in static images.

## 4.1 Implementation Details

We train two variations of our model. One that uses ARFlow [33], and one that uses RAFT [54] as the optical flow network. Since ARFlow is self-supervised, using it results in a fully self-supervised motion magnification method. RAFT on the other hand is a supervised flow model with very good performance, giving us a weakly-supervised magnification method that serves as a rough upper bound to the performance of a fully self-supervised method. More discussion can be found in Section A2 in the appendix. All qualitative results presented in this paper are from the RAFT model, except in Figure 6 in which we compare the ARFlow and RAFT models.

Because motion magnification is multiplicative, we sample random magnification factors $\alpha$ exponentially, such that $\log_2(\alpha) \sim U(\log_2(\alpha_{\min}), \log_2(\alpha_{\max}))$, where $\alpha_{\min} = 1$ and $\alpha_{\max} = 16$. For larger magnification factors that exceed $\alpha_{\max}$, we find that recursive application of our model produces high-quality predictions. More implementation details are available in Section A2 in the appendix.

## 4.2 Dataset

Because our method is self-supervised, we benefit from a large, diverse dataset of videos. To this end we curate a dataset containing 145k unlabeled frame pairs from several existing datasets, including YouTube-VOS-2019 [68], DAVIS [42], Vimeo-90k [66], Tracking Any Object (TAO) [8], and Unidentified Video Objects (UVO) [60]. We remove frame pairs with large motions (e.g., from the camera or objects), since these frames are less likely to be used in magnification appliactions. We also remove frame pairs that are near-identical by setting a lower bound on the MSE between the two frames. In addition to a training set, we collect a test set consisting 650 frame pairs for evaluation, which we refer to as **real-world test set**. We provide more details of dataset collection and filtering in Section A3 of the appendix.

For testing, we also use the synthetic test set from Oh *et al.* [40], which is generated by compositing objects from PASCAL VOC [12] to backgrounds from COCO [30] at varying levels of subpixel motion and noise. We refer to this test set as the **synthetic test set**. Finally, we qualitatively assess our method and baselines on various **real-world videos**, which vary greatly in subject matter, motion complexity, and lighting conditions. These videos include de-facto standard benchmarks for motion magnification used in previous works [64, 58, 59, 5, 40], as well as new videos.

## 4.3 Metrics and Baselines

**Metrics.**   Similar to Oh *et al.* [40], we use **SSIM** [62] as an evaluation metric. However, because SSIM requires ground-truth magnified images, we can only use this metric to evaluate models on the synthetic test set. In order to evaluate our method on the real-world test set, we propose another metric for motion magnification that does not require ground truth, which we term **motion error**. Inspired by the accuracy and robustness of recent optical flow methods, we assume that the flow estimate from an optical flow method is ground truth and calculate an end-point error between the predicted flow and the desired magnified flow.

This metric is identical to $\mathcal{L}_{\mathrm{mag}}$ if the same optical flow model is used for evaluation as is used during training. In order to ensure a robust metric, we calculate the motion error metric using a wide range of optical flow networks, including PWC-Net, GMFlow, and RAFT, that have been trained on various datasets. We set the number of iterations for RAFT to 20 during evaluation, while keeping it at 5 during training. This serves the dual purpose of allowing us to train more efficiently, but also evaluate more equitably.

In addition, we compute the per-pixel ratio between the flow magnitudes of the unmagnified and magnified frames, and calculate their average deviation from the desired magnification factor $\alpha$. This metric, which we refer to as **magnification error**, was not explicitly trained for and serves as another indicator of magnification quality.

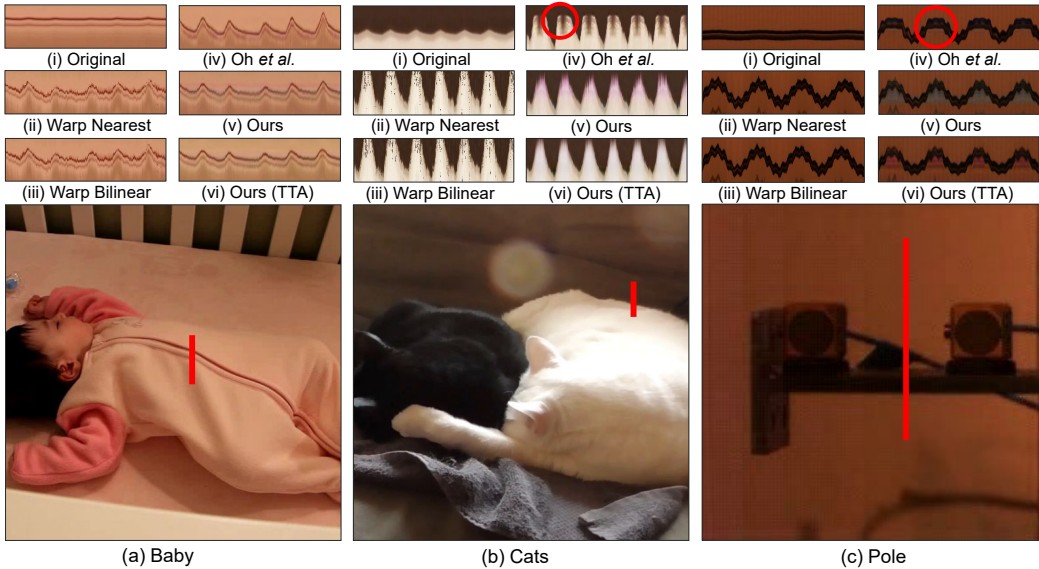

Figure 3: **Qualitative Comparison. (Top)** We visualize the results of various models by plotting $y$-$t$ slices through video volumes. **(Bottom)** We also show the reference frame and the locations of the $y$-$t$ slices. Oh *et al.* (iv) shows noticeable artifacts, indicated by red circles. In (vi) we see that test time adaptation (TTA) can improve our method on out-of-domain inference videos.

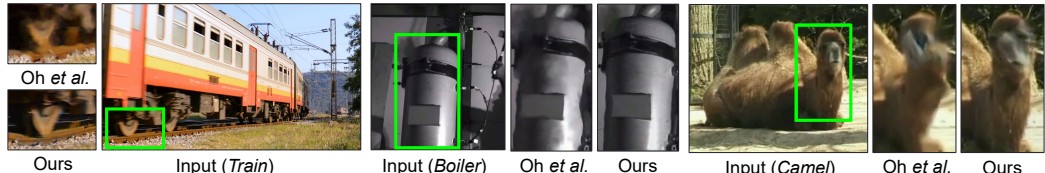

Figure 4: **Image Quality Comparison.** We show magnified frames generated from the method of Oh *et al.* and our method. Because the model of Oh *et al.* is trained on a synthetic dataset, it may not generalize to the presence of novel motions such as the camel chewing, or novel scenes, such as the *train* and *boiler* sequences.

**Baselines.**  We compare primarily against the method of Oh *et al.* [40], a neural network based approach trained on a synthetic dataset. Additionally, we implement forward warp baselines that use nearest and bilinear sampling to warp pixels according to the amplified flow, along with a nonparametric inpainting method [41], which we term **Warp Nearest** and **Warp Bilinear** respectively. We also compare against **FLAVR** [28], a 3D U-Net trained for frame interpolation and finetuned on the same synthetic dataset as Oh *et al.*, albeit at a constant magnification factor of $\alpha = 10$. Finally, we provide qualitative comparisons in Section A5 of the appendix to **Neural Implicit Video Representations (NIVR)** [36], a method that fits an implicit representation to a video and displays emergent motion manipulation behavior.

## 4.4 Comparison with the State-of-the-Art

**Visual quality.**  We show qualitative results on real-world videos. In Figure 4 we show still frames and closeups of results from Oh *et al.* and our method. And in Figure 3 we plot $y$-$t$ slices through the video volume, with one dimension being time and the other being spatial[3].

One qualitative finding is that while the flow network may be noisy on a specific video, our model trained with the same flow network can be much more robust. This can be seen clearly in the *baby* sequence of Figure 3. The forward warp methods, which depend on the optical flow estimate on the inference video, are very jittery whereas our predictions are much smoother. In effect, our method distills a given flow network into a more robust estimator of object motion.

---

[3]For simplicity, we refer both vertical and horizontal slices to $y$-$t$ slices.

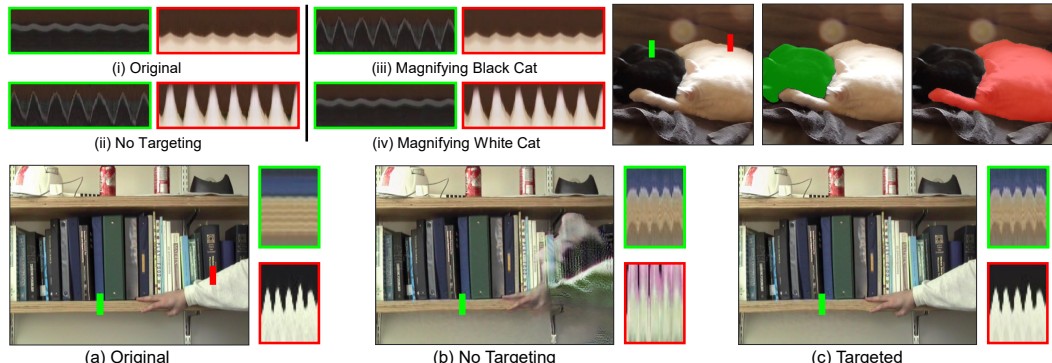

Figure 5: **Targeted Magnification.** Our network is capable of targeting motion magnification. This is useful when we want to focus on a specific object, as in the *cat* sequence above, or when we want to ignore an object that may be challenging to magnify, such as the quickly moving arm in the *bookshelf* sequence above.

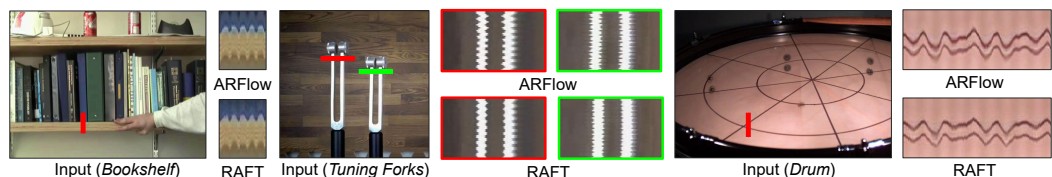

Figure 6: **Qualitative Comparison between Models Trained with ARFlow and RAFT.** We present $y$-$t$ slices from videos magnified by models trained with ARFlow and RAFT. The model (ARFlow) produces magnifications of comparable quality as the model (RAFT), despite being trained on an unsupervised optical flow method.

Another point to note is that the model of Oh *et al.*, trained using synthetic data with piece-wise linear motions, tends to suffer when the motion is complex. For example in Figure 3, with the moving fur in the *cat* sequence, or in Figure 4 with the train tracks. Oh *et al.* also fails when the motion is extreme, as is in the case of the *pole* sequence in Figure 3 where it fails to magnify an object to its full range of motion. Our model on the other hand generalizes to these methods quite well and is further improved on such out-of-domain videos with test time adaptation.

**Targeted magnification.** In Figure 5 we show results of targeted magnification. Given a mask, we can easily magnify just the motion in that mask. This is useful when we want to focus on the motion of a specific object, such as in the *cat* sequence, or when we want to mask out an object whose motion is too large to be magnified, as is the case with the arm in the *bookshelf* sequence.

**ARFlow and RAFT models.** We additionally provide qualitative comparisons between our method trained with the unsupervised, and less powerful, ARFlow optical flow model and our method trained with the supervised RAFT model, in Figure 6. As can be seen, despite making slightly less accurate flow estimates, the ARFlow model is fairly comparable in quality to the RAFT model. This shows that our method can enable a fully self-supervised motion magnification model, in which each component is trained with unlabeled data. For all other figures in the paper, we visualize videos generated by our RAFT model.

**Quantitative evaluation on real-world videos.** We compare our method to baselines on our real-world test set in Table 1 and Figure 7. We magnify frame pairs by magnification factors of $\alpha$ ranging from 1 to 64. Because FLAVR is only trained for $\alpha = 10$, we also evaluate this setting. We achieve the best results on almost all evaluation metrics, even with flow models that we did not train for such as PWC-Net and GMFlow, indicating that we can robustly magnify motions.

**Quantitative evaluation on synthetic videos.** Additionally, we evaluate our method and baselines on the synthetic test sets. The subpixel subset contains frame pairs with 15 levels of purely translational motion varying from 0.04 pixels to 1.0 pixel, and with a fixed target magnified motion magnitude of 10 pixels. The noise subset contains 21 groups of frame pairs with simulated photon noise injected at increasing multiplicative factors, all with a max motion of 2 pixels and fixed magnified motion magnitude of 10 pixels. We report the motion error, magnification error, and SSIM in Table 2 and Figure 8. To compute the motion and magnification error, we use GMFlow [65] trained

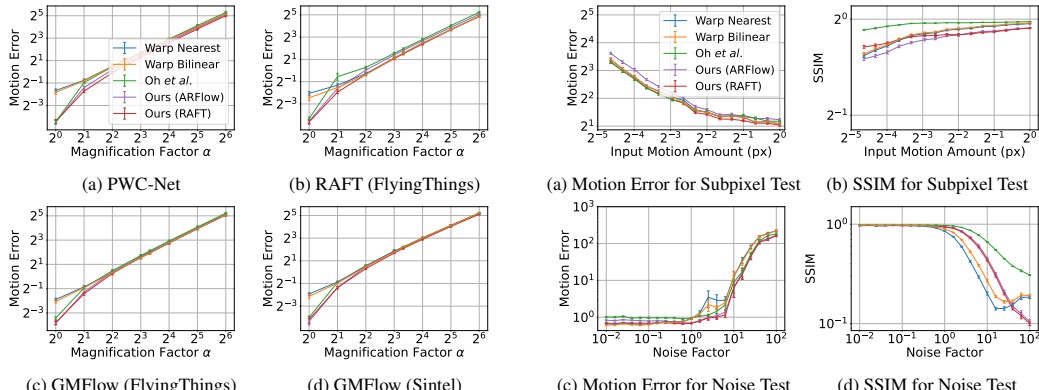

(a) PWC-Net  (b) RAFT (FlyingThings)  (a) Motion Error for Subpixel Test  (b) SSIM for Subpixel Test

(c) GMFlow (FlyingThings)  (d) GMFlow (Sintel)  (c) Motion Error for Noise Test  (d) SSIM for Noise Test

Figure 7: **Evaluation on real-world test set.** We evaluate the methods with $\alpha$ ranging from 1 to 64 and various flow methods. Error bars show the standard error. On motion error, our method consistently obtains more accurate magnified motions.

Figure 8: **Evaluation on synthetic data.** We evaluate our model using the synthetic test dataset from Oh *et al.* [40]. We use two subsets and two metrics (motion error and SSIM). Error bars show the standard error.

Table 1: **Quantitative Evaluation Results on Real-world Test Set.** We report the motion error of our method and baselines on our collected evaluation set. A smaller motion error represents better magnification quality. For fair comparison, we show results for optical flow methods not used during training, including PWC-Net [51] and GMFlow [65] trained with FlyingThings [37]. These results are a subset of those in Figure 7. We achieve better motion magnification on almost all evaluation metrics.

| Method | \multicolumn{5}{c}{PWC-Net} | | | | | \multicolumn{5}{c}{RAFT} | | | | | \multicolumn{5}{c}{GMFlow} | | | | |
|---|---|---|---|---|---|---|---|---|---|---|---|---|---|---|---|
| | $\alpha$=2 | $\alpha$=4 | $\alpha$=10 | $\alpha$=16 | $\alpha$=64 | $\alpha$=2 | $\alpha$=4 | $\alpha$=10 | $\alpha$=16 | $\alpha$=64 | $\alpha$=2 | $\alpha$=4 | $\alpha$=10 | $\alpha$=16 | $\alpha$=64 |
| Warp Nearest | 0.59 | 1.42 | 4.15 | 7.24 | 35.76 | 0.40 | 0.83 | 2.91 | 5.49 | 28.87 | 0.56 | 1.26 | 3.84 | 6.86 | 33.61 |
| Warp Bilinear | 0.57 | 1.40 | 4.12 | 7.28 | 35.78 | 0.35 | **0.78** | 2.97 | 5.52 | **28.70** | 0.53 | 1.24 | 3.79 | 6.88 | 33.72 |
| FLAVR | - | - | 4.24 | - | - | - | - | 3.87 | - | - | - | - | 4.40 | - | - |
| Oh *et al.* | 0.51 | 1.37 | 4.29 | 7.60 | 39.49 | 0.47 | 1.21 | 3.86 | 6.95 | 37.53 | 0.52 | 1.40 | 4.34 | 7.64 | 37.31 |
| Ours (ARFlow) | 0.38 | 1.15 | 3.70 | 6.58 | 34.21 | 0.33 | 1.03 | 3.52 | 6.33 | 32.89 | 0.41 | 1.30 | 4.15 | 7.23 | 35.02 |
| Ours (RAFT) | **0.30** | **0.95** | **3.32** | **6.01** | **32.01** | **0.26** | **0.78** | **2.82** | **5.20** | 29.26 | **0.37** | **1.19** | **3.88** | **6.73** | **33.49** |

Table 2: **Quantitative Evaluation Results on Synthetic Videos.** We report the motion error, magnification error, and SSIM of our method and baselines on two synthetic evaluation sets. Smaller motion error or magnification error, and larger SSIM indicate better quality. We use GMFlow trained with Flying Things to calculate motion error and magnification error. These results are a subset of those plotted in Figure 8.

| | \multicolumn{6}{c}{Motion Error ↓} | | | | | | \multicolumn{6}{c}{Magnification Error ↓} | | | | | | \multicolumn{6}{c}{SSIM ↑} | | | | | |
|---|---|---|---|---|---|---|---|---|---|---|---|---|---|---|---|---|---|---|
| | \multicolumn{3}{c}{Subpixel Test} | | | \multicolumn{3}{c}{Noise Test} | | | \multicolumn{3}{c}{Subpixel Test} | | | \multicolumn{3}{c}{Noise Test} | | | \multicolumn{3}{c}{Subpixel Test} | | | \multicolumn{3}{c}{Noise Test} | | |
| Method | 0.04px | 0.2px | 1px | 0.01x | 1x | 100x | 0.04px | 0.2px | 1px | 0.01x | 1x | 100x | 0.04px | 0.2px | 1px | 0.01x | 1x | 100x |
| Warp Nearest | 10.60 | 3.01 | 2.12 | 0.65 | 0.91 | 221.32 | 209.10 | 22.61 | 3.12 | 0.16 | 0.23 | 1.90 | 0.77 | 0.92 | 0.97 | 0.97 | 0.85 | 0.18 |
| Warp Bilinear | 10.61 | 3.02 | 2.12 | **0.61** | 0.90 | 226.37 | 209.35 | 22.61 | 3.10 | **0.15** | 0.23 | 1.68 | 0.78 | 0.93 | 0.97 | **0.98** | 0.90 | 0.19 |
| Oh *et al.* | **9.82** | 2.93 | 2.26 | 1.01 | 0.94 | 181.26 | 193.78 | 22.05 | 3.15 | 0.28 | 0.25 | 1.70 | **0.93** | **0.97** | **0.98** | 0.98 | **0.97** | **0.31** |
| Ours (ARFlow) | 12.29 | 3.26 | 2.35 | 0.82 | 0.70 | 167.77 | 222.16 | 20.87 | 3.05 | 0.20 | 0.18 | 1.69 | 0.75 | 0.89 | 0.94 | 0.97 | 0.94 | 0.10 |
| Ours (RAFT) | 10.02 | **2.80** | **2.05** | 0.69 | **0.68** | **164.36** | **175.45** | **17.01** | **2.48** | 0.17 | **0.17** | **1.49** | 0.82 | 0.90 | 0.94 | 0.97 | 0.94 | 0.10 |

with FlyingThings [37]. We achieve satisfactory results on the motion error metrics and lag slightly behind on SSIM.

# 5 Discussion

We present a method for learning to magnify subtle motions through self-supervised learning. We demonstrated its effectiveness through a variety of experiments, using quantitative evaluations on real and synthetic data, and through qualitative results on videos containing complex motions. We also showed that the flexibility of our model allowed it to be extended through test-time adaptation and targeted motion magnification. We see our work as opening new directions in visualizing subtle motions. First, while we have shown one method for defining the loss function, based on optical flow, our approach could be applied to other differentiable motion estimation techniques, such as emerging

methods for long-range tracking [9, 22, 2]. Second, the flexibility of our model opens the possibility for other "user in the loop" extensions, beyond allowing for segmentation-based magnification.

**Broader impacts.** Motion magnification methods have a range of applications, such as amplifying subtle motions in biology and engineering [57], amplifying microexpressions for assistive technology [63], assisting measurement techniques [41], and for video forensics [6, 7]. Please see our website for potential applications. It also has the potential negative use of revealing body motions that a person may think are undetectable, such as one's pulse [64, 1], which may impinge on privacy.

**Limitations.** The performance of our model is closely tied to the limitations and capabilities of the underlying optical flow estimator. We share the limitation with existing work [40] that our method works by magnifying motion between the first frame and every subsequent frame, which may pose challenges in the presence of occlusion and disocclusion. The magnification loss (Eq. 3) may still be applicable in these cases, since modern optical flow models are trained to track occluded pixels. However, the photoconsistency assumption (Eq. 4) will no longer hold. For simplicity, we do not inpaint missing image regions [32], and we use distance between pixels in lieu of a perceptual loss or a generative model. This simplicity has potential advantages for visualization applications, since it avoids hallucinating scene structure that is not present in the original images, but it also sometimes results in undesirable artifacts.

**Acknowledgements.** We thank Tae-Hyun Oh for the extensive discussions and data. We also thank Byung-Ki Kwon, Tarun Kalluri, Long Mai, and Stella Yu for helpful discussions. Daniel Geng is supported by a National Science Foundation Graduate Research Fellowship under Grant No. 1841052. This work was supported in part by DARPA Semafor, Program No. HR001120C0123. The views, opinions and/or findings expressed are those of the authors and should not be interpreted as representing the official views or policies of the Department of Defense or the U.S. Government.

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

## A1 Video Results

We highly encourage the reader to visit our webpage to see qualitative results and comparisons against other methods. Results on the website include:

- Videos magnified at various magnification factors, and side-by-side comparisons against Oh *et al.* [40], FLAVR [28] and NIVR [36].
- Targeted magnification results.
- Test-time adaptation results.
- Failure cases (see Section A8).
- Ablations with no regularization (see Section A6).

## A2 Training Details

**Architecture.** We use a U-Net [45] that takes in 2 frames as input (concatenated channel-wise) and a positional encoding for $\alpha$ (described below) and outputs a single frame. Our U-Net is composed of 5 layers of downsampling and 5 layers of upsampling, with skip connections between corresponding layers during downsampling and upsampling. We use a filter size of 64 at the shallowest layer, with the number of filters doubling at each downsampling, ending with a filter size of 1024 for the coarsest layer. Downsampling is performed via max pooling with a kernel size $2 \times 2$ and stride 2, and the upsampling is performed using bilinear upsampling. Each layer is composed of a convolution of kernel size $3 \times 3$, batch normalization, and a ReLU activation, all applied twice. We apply a sigmoid activation at the very end to transform the final output into the range $(0, 1)$. Our model has a total of 17.3M trainable parameters, and the model of Oh *et al.* has 0.92M parameters. Note that the model of Oh *et al.* uses a bespoke architecture designed specifically for motion magnification, consisting of an encoder, a decoder, and a manipulator, and is fundamentally different from ours.

To encode magnification factors for input, we transform a single scalar $\alpha$ indicating the desired magnification factor to a 32 dimensional vector. We use sinusoidal positional embeddings with frequencies ranging geometrically from $2^{-3}$ to $2^7$. In order to feed the embeddings into the U-Net we tile it spatially and concatenate it with the two input frames. The resulting input to the U-Net has a channel size of 38, where 6 come from the two input frames and 32 come from the encoding. At training time the spatially tiled positional embeddings are constant—they all represent the same $\alpha$ value. However, at test time it is possible for the embeddings to vary spatially, allowing for various magnification factors in different areas of the video, and enabling targeted magnification.

**Training parameters.** The parameter $\lambda_{\text{color}}$ is set to 10, while the magnification factors $\alpha$ range from 1 to 16, sampled geometrically. The learning rate is $3 \times 10^{-4}$, and model training is performed using a batch size of 40 with 4 A40 GPUs and an image size of $512 \times 512$. In terms of data augmentation, a random area is initially cropped with a scale within the range of $(0.7, 1.0)$. The cropped area is then resized to dimensions of $512 \times 512$. Furthermore, the image is subjected to random horizontal or vertical flipping with a probability of 0.5, as well as random rotation within the range of $(-15°, 15°)$. Finally, strong color jittering is applied to the frames. These transformations are applied identically to both frames. For the optical flow model, we select ARFlow [33] pretrained with KITTI 2015 [38] and RAFT [54] pretrained on FlyingThings [37]. We use 5 iterations of the RAFT model to compute the flows required by our losses.

**Choice of optical flow models.** We implemented our algorithm using two distinct optical flow models, namely RAFT [54] and ARFlow [33]. During the training process, we kept the pretrained optical flow model weights fixed, employing them solely to provide motion estimates for our loss computations and gradients for backpropagation to train the U-Net. We use ARFlow, an unsupervised model, to demonstrate that our method works in a truly self-supervised manner, in which no component is trained with labeled data. We then use RAFT, a strong supervised flow network, to train a high quality, weakly-supervised magnification model that serves as a rough upper bound to the performance of a fully self-supervised model. We note that the comparison of quantitative results of both models indicates that our method has better performance with better optical flow quality, hinting that our framework can be benefit from future improvement of optical flow models.

Table A1: **Dataset Statistics (Train Set).** We report basic statistics of our training dataset, including the temporal sampling stride and the number of frame pairs before and after filtering.

| | YouTube-VOS | DAVIS | Vimeo-90k | TAO | | UVO | | Total |
|---|---|---|---|---|---|---|---|---|
| Sampling Stride | 1 | 1 | 1 | 1 | 5 | 1 | 5 | - |
| # Raw Pairs | 46,858 | 3,089 | 193,836 | 176,092 | 175,434 | 71,836 | 70,830 | 737,975 |
| # Seleted Pairs | 4,050 | 370 | 28,734 | 50,441 | 29,795 | 22,462 | 9,749 | 145,601 |

## A3  Dataset Details

**Dataset overview.**    To train and evaluate our model, we compile a large dataset of frame pairs by sampling from various existing datasets: Youtube-VOS-2019 [68], DAVIS [42], Vimeo-90k [66], Tracking Any Object (TAO) [8], and Unidentified Video Objects (UVO) [60].The train set is collected by sampling from the train sets of the above datasets, with the exception of DAVIS, where we use the trainval split due to its smaller size. To construct a real-world test set, a total of 50 frame pairs are randomly collected from each dataset's test set, except for the TAO dataset, in which the validation set is used. Because the TAO dataset is comprised of five distinct datasets (ArgoVerse [4], BDD-100K [69], Charades [46], LaSOT [13], and YFCC100M [56]), we sample 50 frame pairs from each of these sub-datasets. In total, 650 frame pairs are acquired across all datasets for evaluation purposes. Basic statistics of the collected dataset is listed in Table A1.

**Sampling and filtering.**    We use consistent filtering and sampling approaches for both the training dataset and the real-world test set. We use two sampling methods, characterized by different temporal strides (1 or 5) indicating the temporal distance between two sampled frames. Specifically, a stride of 5 means that frame $i + 5$ is sampled to form a frame pair alongside frame $i$. When using a stride of 1, non-overlapping consecutive frame pairs are sampled. For the datasets Youtube-VOS-2019, DAVIS, and Vimeo-90k, a stride of 1 is used. In the case of TAO and UVO, strides of 1 and 5 are utilized to supplement the dataset with additional samples due to the high framerates in these datasets.

After obtaining raw frame pairs from these datasets, a filtering process is conducted to eliminate frame pairs exhibiting significant object or camera motion. We estimate motion using RAFT and set upper bounds for different quantiles of the per-pixel flow magnitude distribution. These include an upper bound of 20 on the 99.9th percentile, an upper bound of 2 on the 80th percentile, and an upper bound of 0.1 on the 0.01st percentile. We find empirically that these thresholds result in frame pairs that have little camera motion and small object motions. To filter out identical or near-identical frame pairs, an additional lower bound is introduced to filter out frame pairs with a mean squared error (MSE) of less than 10 between the two frames.

After sampling and filtering, we obtain a training dataset of 145k frame pairs. Detailed information regarding the number of sampled raw pairs and the number of selected pairs can be found in Table A1 in our paper.

## A4  Figure Parameters

We give a comprehensive summary of the magnification factors used and location of $y$-$t$ slices or closeups for each sequence in Table A2 and Table A3. Additionally, we also provide the corresponding figure index in our paper, and the frame index, indicating which frame we displayed in our figures. Note that some images are cropped or stretched to optimize their presentation within the figures.

## A5  Additional Qualitative Results

**Comparison with NIVR.**    Neural implicit video representations (NIVR) [36] is a method that encodes a video into a neural implicit representation. Additionally, the method uses a phase-based positional encoding which allows for control over motion. However, the method does not give precise control over magnification factors as the manipulation is over the latent positional encodings, and is only shown to perform motion magnification up to a factor of about 2 (our work and previous works can achieve factors of up to 200). Furthermore, the code for their method is unreleased. Therefore we compare our method against their results obtained through correspondence with the authors. We show $y$-$t$ slices in Figure A1. We achieve comparable performance on $y$-$t$ slices for the *flower* and *guitar*

Table A2: **Information on $y$-$t$ Slices of Magnified Sequences.** We represent the location of $y$-$t$ slices with the upper left $(x_1, y_1)$ coordinate and bottom right coordinate $(x_2, y_2)$ for every sequence appeared in our paper. We also provide magnification factors for each $y$-$t$ slices. For frames with two $y$-$t$ slices in the same frame, we report the left and right locations accordingly, and mark with "L" for left, "R" for right in the table. For sequences magnified by different magnification factors, we report information for every motion magnification factor.

| Sequence | *Drum* | *Pole* | *Tuning Forks* (L) | *Tuning Forks* (R) | *Baby* | *Cats* (L) |
|---|---|---|---|---|---|---|
| Figure | Figure 1,6 | Figure 1, 3 | Figure 1, 6, A3 | Figure 1, 6, A3 | Figure 3 | Figure 5 |
| $\alpha$ | 5 | 20 | 5 | 5 | 20 | 20 |
| Upper Left | (180, 250) | (20, 120) | (175, 200) | (340, 260) | (510, 212) | (280, 160) |
| Bottom Right | (181, 310) | (140, 121) | (315, 201) | (480, 261) | (511, 292) | (281, 300) |

| Sequence | *Cats* (R) | *Bookshelf* (L) | *Bookshelf* (R) | *Flower* | *Guitar1* | *Camel* |
|---|---|---|---|---|---|---|
| Figure | Figure 3,5 | Figure 5,6 | Figure 5 | Figure A1 | Figure A1 | Figure A1 |
| $\alpha$ | 20 | 15 | 15 | 2 | 2 | 2 |
| Upper Left | (710, 130) | (240, 340) | (560, 260) | (260, 50) | (20, 120) | (250, 90) |
| Bottom Right | (711, 270) | (241, 400) | (561, 320) | (350, 51) | (21, 160) | (280, 91) |

| Sequence | *Cats* (R) | *Bookshelf* (L) | *Boiler* | *Flower* | *Guitar2* | |
|---|---|---|---|---|---|---|
| Figure | Figure A2 | Figure A2 | Figure A3,A4 | Figure A5 | Figure A5 | |
| $\alpha$ | 10 | 10 | 30 | 20 | 25 | |
| Upper Left | (710, 130) | (240, 340) | (100, 200) | (240, 10) | (280, 50) | |
| Bottom Right | (711, 270) | (241, 400) | (200, 201) | (241, 50) | (281, 100) | |

Table A3: **Information on Closeups of Magnified Sequences.** We represent the location of closeups with the upper left $(x_1, y_1)$ coordinate and bottom right coordinate $(x_2, y_2)$. We include frame index that appears in our figures, and frame index starts from 1 for a video. We also provide magnification factors for each closeups in our paper.

| Sequence | *Train* | *Pole* | *Train* | *Camel* | *Boiler* |
|---|---|---|---|---|---|
| Figure | Figure 1 | Figure 1 | Figure 4 | Figure 4 | Figure 4 |
| $\alpha$ | 20 | 20 | 20 | 20 | 30 |
| Upper Left | (20, 520) | (0, 0) | (50, 520) | (280, 50) | (100, 120) |
| Bottom Right | (300, 710) | (200, 200) | (240, 710) | (420, 300) | (350, 576) |
| Frame Index | 417 | 1, 190 | 417 | 30 | 19 |

| Sequence | *Train* | *Tuning Forks* | *Boiler* | *Boiler* | *Camera* |
|---|---|---|---|---|---|
| Figure | Figure A2 | Figure A3 | Figure A3 | Figure A4 | Figure A5 |
| $\alpha$ | 10 | 5 | 30 | 30 | 75 |
| Upper Left | (50, 520) | - | - | (80, 120) | (0, 0) |
| Bottom Right | (240, 710) | - | - | (280, 320) | (45, 200) |
| Frame Index | 417 | 10 | 19 | 19 | 16 |

Table A4: **Quantitative Evaluation Results on Real-world Videos.** We report full evaluation results on real-world videos. Performance is measured by motion error and magnification error for various optical flow estimators.

| | | Motion Error ↓ | | | | | | | Magnification Error ↓ | | | | | | |
|---|---|---|---|---|---|---|---|---|---|---|---|---|---|---|---|
| | | $\alpha$=1 | $\alpha$=2 | $\alpha$=4 | $\alpha$=8 | $\alpha$=16 | $\alpha$=32 | $\alpha$=64 | $\alpha$=1 | $\alpha$=2 | $\alpha$=4 | $\alpha$=8 | $\alpha$=16 | $\alpha$=32 | $\alpha$=64 |
| PWC-Net | Warp Nearest | 0.32 | 0.59 | 1.42 | 3.16 | 7.24 | 16.24 | 35.76 | 0.56 | 1.07 | 2.28 | 4.65 | 9.54 | 19.94 | 43.51 |
| | Warp Bilinear | 0.28 | 0.57 | 1.40 | 3.16 | 7.28 | 16.18 | 35.78 | 0.63 | 1.08 | 2.21 | 4.51 | 9.45 | 19.82 | 43.37 |
| | Oh *et al.* | 0.05 | 0.51 | 1.37 | 3.27 | 7.60 | 17.52 | 39.49 | 0.20 | 0.80 | 2.06 | 4.51 | 9.65 | 21.58 | 50.36 |
| | Ours (ARFlow) | **0.04** | 0.38 | 1.15 | 2.81 | 6.58 | 15.18 | 34.21 | **0.13** | 0.74 | 2.13 | 4.52 | 9.19 | 19.68 | 41.51 |
| | Ours (RAFT) | 0.05 | **0.30** | **0.95** | **2.48** | **6.01** | **14.05** | **32.01** | **0.13** | **0.55** | **1.56** | **3.62** | **7.74** | **16.22** | **34.82** |
| RAFT | Warp Nearest | 0.24 | 0.40 | 0.83 | 2.19 | 5.49 | 12.71 | 28.87 | 0.65 | 1.18 | 2.25 | 4.33 | 8.79 | 18.04 | 39.63 |
| | Warp Bilinear | 0.18 | 0.35 | 0.78 | 2.20 | 5.52 | 12.81 | **28.70** | 0.76 | 1.15 | 2.08 | 4.46 | 9.01 | 17.90 | 38.22 |
| | Oh *et al.* | 0.05 | 0.48 | 1.21 | 2.92 | 6.94 | 16.39 | 37.55 | 0.29 | 0.83 | 1.90 | 4.20 | 9.33 | 21.44 | 51.07 |
| | Ours (ARFlow) | **0.04** | 0.33 | 1.03 | 2.64 | 6.33 | 14.63 | 32.89 | 0.19 | 0.75 | 2.06 | 4.68 | 10.25 | 22.51 | 47.28 |
| | Ours (RAFT) | **0.04** | **0.26** | **0.78** | **2.10** | **5.20** | **12.66** | 29.26 | **0.16** | **0.56** | **1.58** | **3.79** | **8.34** | **17.49** | **35.59** |
| GMFlow (FlyingThings) | Warp Nearest | 0.27 | 0.56 | 1.26 | 2.93 | 6.86 | 15.43 | 33.61 | 0.61 | 1.20 | 2.52 | 5.14 | 10.43 | 21.17 | 44.56 |
| | Warp Bilinear | 0.24 | 0.53 | 1.24 | 2.94 | 6.88 | 15.46 | 33.72 | 0.64 | 1.17 | 2.42 | 5.05 | 10.38 | 21.44 | 45.14 |
| | Oh *et al.* | 0.09 | 0.52 | 1.40 | 3.31 | 7.64 | 16.84 | 37.31 | 0.47 | 0.92 | 2.11 | 4.75 | 10.53 | 22.74 | 51.99 |
| | Ours (ARFlow) | **0.07** | 0.41 | 1.30 | 3.18 | 7.23 | 16.68 | 35.02 | 0.28 | 0.82 | 2.18 | 4.64 | 9.72 | 21.47 | 44.74 |
| | Ours (RAFT) | **0.07** | **0.37** | **1.19** | **2.98** | **6.73** | **15.32** | **33.49** | 0.26 | **0.65** | **1.81** | **4.36** | **9.11** | **19.62** | **42.17** |
| GMFlow (Sintel) | Warp Nearest | 0.26 | 0.55 | 1.43 | 3.64 | 8.02 | 17.32 | 36.69 | 0.66 | 1.36 | 3.25 | 7.73 | 16.76 | 33.61 | 66.49 |
| | Warp Bilinear | 0.23 | 0.51 | 1.39 | 3.45 | 8.04 | 17.19 | 36.84 | 0.96 | 1.70 | 3.57 | 7.79 | 16.64 | 33.53 | 66.57 |
| | Oh *et al.* | 0.06 | 0.53 | 1.49 | 3.60 | 8.13 | 17.53 | 37.31 | 0.65 | 1.09 | 2.40 | 5.26 | 11.28 | 24.01 | 51.71 |
| | Ours (ARFlow) | **0.05** | **0.39** | 1.38 | 3.49 | 7.87 | 17.25 | 36.10 | **0.26** | 0.96 | 2.91 | 6.45 | 13.27 | 28.34 | 55.73 |
| | Ours (RAFT) | 0.06 | **0.39** | **1.24** | **3.22** | **7.43** | **16.33** | **34.44** | 0.28 | **0.72** | **2.03** | **4.94** | **10.88** | **23.45** | **47.67** |

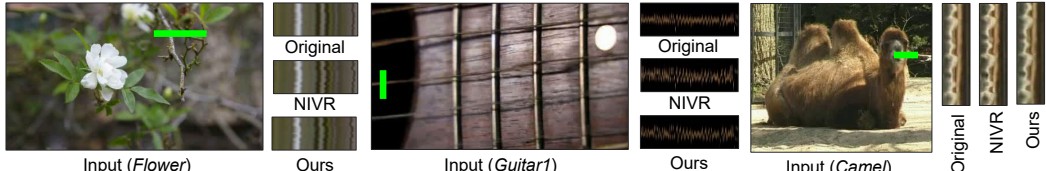

Figure A1: **Qualitative Comparison with NIVR.** NIVR [36] magnifies a learned implicit representation, which does not map cleanly to a precise magnification factor. We found using $\alpha = 2$ with our method gave comparable magnification to the results from NIVR. Because there is no publicly released code for NIVR, we compare against videos and results from the NIVR website. Above we visualize the $y\text{-}t$ slices of the original videos, NIVR videos, and our videos for comparison.

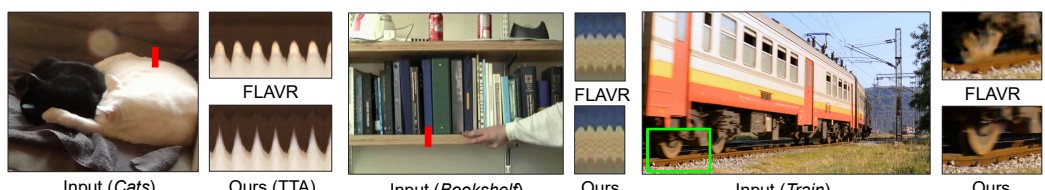

Figure A2: **Qualitative Comparison with FLAVR.** We use $\alpha = 10$ for our method and compare the results with the magnified videos of FLAVR. We show the $y\text{-}t$ slices of FLAVR videos and our videos. We also show a closeup in the *train* sequence, shown in the green rectangle.

Table A5: **Quantitative Evaluation Results on Synthetic Videos.** We report the evaluation results on the sub-pixel test with synthetic videos with all different input motion amount, with three metrics including motion error, magnification error, and SSIM.

| Motion Error ↓ | | | | | | | | | | | | | | |
|---|---|---|---|---|---|---|---|---|---|---|---|---|---|---|
| Input Motion | 0.04px | 0.05px | 0.06px | 0.08px | 0.10px | 0.13px | 0.16px | 0.20px | 0.25px | 0.32px | 0.40px | 0.50px | 0.63px | 0.79px | 1.00px |
| Warp Nearest | 10.60 | 8.34 | 6.80 | 5.45 | 4.77 | 4.13 | 3.80 | 3.01 | 2.80 | 2.56 | 2.57 | 2.49 | 2.21 | 2.24 | 2.12 |
| Warp Bilinear | 10.61 | 8.34 | 6.80 | 5.44 | 4.77 | 4.13 | 3.58 | 3.02 | 2.80 | 2.56 | 2.57 | 2.49 | 2.20 | 2.23 | 2.12 |
| Oh *et al.* | **9.82** | **7.93** | **6.46** | **5.17** | **4.44** | **3.85** | 3.60 | 2.93 | 2.87 | 2.51 | 2.57 | 2.61 | 2.44 | 2.26 | 2.26 |
| Ours (ARFlow) | 12.29 | 9.86 | 8.16 | 6.38 | 5.37 | 4.48 | 3.99 | 3.26 | 3.02 | 2.65 | 2.69 | 2.53 | 2.42 | 2.42 | 2.35 |
| Ours (RAFT) | 10.02 | 7.96 | 6.55 | 5.22 | 4.50 | 3.86 | **3.55** | **2.80** | **2.68** | **2.39** | **2.38** | **2.29** | **2.16** | **2.15** | **2.05** |

| Magnification Error ↓ | | | | | | | | | | | | | | |
|---|---|---|---|---|---|---|---|---|---|---|---|---|---|---|---|
| Input Motion | 0.04px | 0.05px | 0.06px | 0.08px | 0.10px | 0.13px | 0.16px | 0.20px | 0.25px | 0.32px | 0.40px | 0.50px | 0.63px | 0.79px | 1.00px |
| Warp Nearest | 209.10 | 155.27 | 117.06 | 85.36 | 62.23 | 45.11 | 32.52 | 22.61 | 16.61 | 12.75 | 10.71 | 7.98 | 5.29 | 4.59 | 3.12 |
| Warp Bilinear | 209.35 | 153.32 | 116.98 | 85.40 | 62.24 | 45.37 | 30.09 | 22.61 | 16.55 | 12.73 | 10.70 | 7.99 | 5.29 | 4.58 | 3.10 |
| Oh *et al.* | 193.78 | 145.86 | 106.22 | 79.79 | 56.42 | 42.23 | 30.96 | 22.05 | 17.37 | 12.50 | 10.53 | 7.90 | 5.55 | 4.39 | 3.15 |
| Ours (ARFlow) | 222.16 | 165.24 | 122.15 | 86.58 | 62.42 | 42.24 | 29.68 | 20.87 | 15.86 | 11.45 | 8.78 | 6.67 | 5.03 | 3.99 | 3.05 |
| Ours (RAFT) | **175.45** | **133.33** | **97.06** | **69.86** | **52.10** | **36.56** | **25.42** | **17.01** | **13.36** | **9.86** | **8.23** | **5.94** | **4.20** | **3.32** | **2.48** |

| SSIM ↑ | | | | | | | | | | | | | | |
|---|---|---|---|---|---|---|---|---|---|---|---|---|---|---|---|
| Input Motion | 0.04px | 0.05px | 0.06px | 0.08px | 0.10px | 0.13px | 0.16px | 0.20px | 0.25px | 0.32px | 0.40px | 0.50px | 0.63px | 0.79px | 1.00px |
| Warp Nearest | 0.77 | 0.80 | 0.82 | 0.86 | 0.89 | 0.90 | 0.91 | 0.92 | 0.93 | 0.94 | 0.95 | 0.95 | 0.96 | 0.96 | 0.97 |
| Warp Bilinear | 0.78 | 0.81 | 0.83 | 0.86 | 0.89 | 0.91 | 0.91 | 0.93 | 0.94 | 0.94 | 0.95 | 0.95 | 0.96 | 0.97 | 0.97 |
| Oh *et al.* | **0.93** | **0.94** | **0.95** | **0.96** | **0.97** | **0.97** | **0.97** | **0.97** | **0.97** | **0.97** | **0.98** | **0.98** | **0.98** | **0.98** | **0.98** |
| Ours (ARFlow) | 0.75 | 0.77 | 0.79 | 0.82 | 0.84 | 0.86 | 0.87 | 0.89 | 0.89 | 0.90 | 0.92 | 0.92 | 0.93 | 0.93 | 0.94 |
| Ours (RAFT) | 0.82 | 0.83 | 0.84 | 0.86 | 0.88 | 0.89 | 0.89 | 0.90 | 0.89 | 0.90 | 0.91 | 0.91 | 0.92 | 0.93 | 0.94 |

sequences, and our $y\text{-}t$ slice for the *camel* sequence is crisper than NIVR. Comparison of videos is available on our webpage.

**Comparison with FLAVR.** We also compare against FLAVR [28], a 3-D U-Net proposed for frame interpolation and downstream tasks including motion magnification. Following correspondence with the authors, we replicate their experiments and finetune the public frame interpolation FLAVR checkpoint for motion magnification by training on a subset of the synthetic dataset of Oh *et al.* [40] with $\alpha \approx 10$. The results of this comparison are displayed in Figure A2. Our magnification has crisper boundary and clearer visual quality. Comparison of videos is available on our webpage.

## A6 Additional Experiments

**Full evaluation results.** In our main paper, due to space limitations we presented selected groups of evaluation results in tables. We present all evaluation results in the following tables for reference,

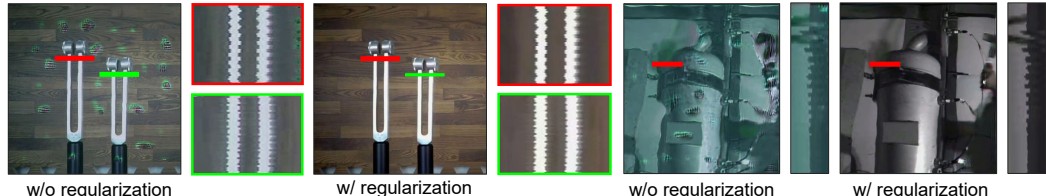

<p style="text-align:center">w/o regularization      w/ regularization      w/o regularization      w/ regularization</p>

Figure A3: **Qualitative Comparison with No Regularization Experiments.** We show magnified frames and $y$-$t$ slices from our model with and without the color loss regularization.

Table A6: **Quantitative Evaluation Results on Synthetic Videos.** We report the evaluation results on the noise test with synthetic videos with all different noise factors, with three metrics including motion error, magnification error, and SSIM.

| Motion Error ↓ | | | | | | | | | | | | | | | | | | | | |
|---|---|---|---|---|---|---|---|---|---|---|---|---|---|---|---|---|---|---|---|---|
| Noise Factor | 0.01 | 0.02 | 0.03 | 0.04 | 0.06 | 0.10 | 0.16 | 0.25 | 0.40 | 0.63 | 1.00 | 1.59 | 2.51 | 3.98 | 6.31 | 10.00 | 15.85 | 25.12 | 39.81 | 63.10 | 100.00 |
| Warp Nearest | 0.65 | 0.66 | 0.69 | 0.66 | 0.64 | 0.65 | 0.71 | 0.75 | 0.76 | 0.76 | 0.91 | 1.30 | 3.49 | 2.87 | 2.85 | 12.43 | 35.57 | 82.03 | 155.00 | 190.06 | 221.32 |
| Warp Bilinear | **0.61** | **0.62** | **0.66** | **0.62** | **0.61** | **0.63** | **0.68** | 0.73 | 0.73 | 0.74 | 0.90 | 1.22 | 2.21 | 1.84 | 2.51 | 12.85 | 31.72 | 78.00 | 153.10 | 182.60 | 226.37 |
| Oh *et al.* | 1.01 | 1.00 | 1.05 | 0.94 | 0.97 | 0.96 | 0.96 | 0.96 | 0.96 | 0.90 | 0.94 | 1.07 | 1.14 | 1.45 | 2.18 | 8.92 | 17.20 | 51.44 | 119.34 | 163.99 | 181.26 |
| Ours (ARFlow) | 0.82 | 0.81 | 0.85 | 0.82 | 0.81 | 0.79 | 0.80 | 0.79 | 0.74 | 0.70 | 0.70 | 0.80 | 1.01 | 1.09 | 1.38 | 6.44 | 16.08 | 46.03 | 115.04 | 135.26 | 167.77 |
| Ours (RAFT) | 0.69 | 0.67 | 0.71 | 0.69 | 0.69 | 0.68 | 0.70 | **0.71** | **0.68** | **0.66** | **0.68** | **0.79** | **0.95** | **1.00** | **1.15** | **6.32** | **14.49** | **44.04** | **107.10** | **128.67** | **164.36** |

| Magnification Error ↓ | | | | | | | | | | | | | | | | | | | | |
|---|---|---|---|---|---|---|---|---|---|---|---|---|---|---|---|---|---|---|---|---|---|
| Noise Factor | 0.01 | 0.02 | 0.03 | 0.04 | 0.06 | 0.10 | 0.16 | 0.25 | 0.40 | 0.63 | 1.00 | 1.59 | 2.51 | 3.98 | 6.31 | 10.00 | 15.85 | 25.12 | 39.81 | 63.10 | 100.00 |
| Warp Nearest | 0.16 | **0.17** | 0.16 | 0.16 | 0.15 | **0.16** | 0.18 | **0.17** | 0.20 | 0.20 | 0.23 | 0.42 | 0.93 | 1.13 | 0.84 | 2.22 | 6.51 | 11.34 | 9.65 | 2.74 | 1.90 |
| Warp Bilinear | **0.15** | 0.17 | **0.16** | **0.15** | **0.14** | **0.16** | 0.17 | **0.17** | 0.19 | 0.19 | 0.23 | 0.39 | 0.58 | 0.51 | 0.74 | 2.21 | 6.19 | 9.40 | 8.47 | 2.68 | 1.68 |
| Oh *et al.* | 0.28 | 0.30 | 0.29 | 0.25 | 0.26 | 0.26 | 0.26 | 0.25 | 0.26 | 0.24 | 0.25 | 0.30 | 0.31 | 0.37 | 0.54 | 1.10 | 1.72 | 4.00 | 3.45 | 1.71 | 1.70 |
| Ours (ARFlow) | 0.20 | 0.20 | 0.22 | 0.20 | 0.20 | 0.19 | 0.20 | 0.19 | 0.19 | 0.18 | 0.18 | 0.22 | 0.24 | 0.27 | 0.34 | **0.71** | 1.48 | **1.97** | 2.67 | 1.65 | 1.69 |
| Ours (RAFT) | 0.17 | **0.17** | 0.18 | 0.17 | 0.17 | 0.17 | **0.17** | **0.17** | **0.17** | 0.16 | **0.17** | **0.21** | **0.22** | **0.24** | **0.31** | 0.81 | **1.32** | 2.28 | **1.70** | **1.50** | **1.49** |

| SSIM ↑ | | | | | | | | | | | | | | | | | | | | |
|---|---|---|---|---|---|---|---|---|---|---|---|---|---|---|---|---|---|---|---|---|---|
| Noise Factor | 0.01 | 0.02 | 0.03 | 0.04 | 0.06 | 0.10 | 0.16 | 0.25 | 0.40 | 0.63 | 1.00 | 1.59 | 2.51 | 3.98 | 6.31 | 10.00 | 15.85 | 25.12 | 39.81 | 63.10 | 100.00 |
| Warp Nearest | 0.97 | 0.97 | 0.97 | 0.97 | 0.97 | 0.97 | 0.97 | 0.96 | 0.95 | 0.92 | 0.85 | 0.75 | 0.59 | 0.42 | 0.29 | 0.20 | 0.14 | 0.14 | 0.15 | 0.18 | 0.18 |
| Warp Bilinear | **0.98** | **0.98** | 0.97 | 0.97 | **0.98** | **0.98** | 0.97 | 0.97 | 0.96 | 0.94 | 0.90 | 0.83 | 0.70 | 0.53 | 0.39 | 0.27 | 0.19 | 0.17 | 0.17 | 0.19 | 0.19 |
| Oh *et al.* | **0.98** | **0.98** | **0.98** | **0.98** | **0.98** | **0.98** | **0.98** | **0.98** | **0.98** | **0.98** | **0.97** | **0.96** | **0.93** | **0.87** | **0.78** | **0.66** | **0.55** | **0.45** | **0.38** | **0.34** | **0.31** |
| Ours (ARFlow) | 0.97 | 0.97 | 0.96 | 0.96 | 0.97 | 0.97 | 0.97 | 0.97 | 0.96 | 0.96 | 0.94 | 0.92 | 0.86 | 0.74 | 0.61 | 0.46 | 0.32 | 0.21 | 0.15 | 0.13 | 0.10 |
| Ours (RAFT) | 0.97 | 0.97 | 0.97 | 0.97 | 0.97 | 0.97 | 0.97 | 0.97 | 0.96 | 0.96 | 0.94 | 0.91 | 0.84 | 0.72 | 0.59 | 0.43 | 0.30 | 0.20 | 0.14 | 0.12 | 0.10 |

including results on real-world videos with different optical flow methods in Table A4 and on all subsets of the synthetic data from Oh *et al.* [40] in Table A5 and Table A6.

**No regularization.** To demonstrate the significance of the color loss, $\mathcal{L}_{\text{color}}$, we ablate it out and show qualitative results for comparison. In Figure A3, we compare our method with and without a color loss with frames from the magnified videos. Notably, we observe that although the magnified motion appears similar to our method with regularization, the color of the results obtained with $\lambda_{\text{color}} = 0$ significantly deviates from the correct colors and has many artifacts.

## A7    Forward Warp with Inpainting

**Implementation.** We introduced two forward warp techniques, which we refer to as *Warp Nearest* and *Warp Bilinear*, as the Lagrangian magnification baselines. To implement these methods, we initially computed optical flow using the same model applied in our experiments (RAFT trained with FlyingThings). Subsequently, we employed forward warping in either a nearest or bilinear manner, applying optical flow values scaled by magnification factors to relocate pixels within the image. Furthermore, we utilized nonparametric inpainting with the built-in functions from OpenCV to address areas left blank due to pixel relocation.

In addition to using nonparametric inpainting, we explored the option of inpainting with DeepFillv2, a powerful learning-based inpainting method proposed by Yu *et al.* [70]. Two different inpainting methods imply that inpainting is not the bottleneck for Lagrangian magnification without deep learning, thus highlighting the advantage of our method. Our evaluation results indicate comparable performance between forward warp techniques with different inpainting methods. Since the forward warp method employing nonparametric inpainting yielded slightly better qualitative and quantitative results, we used forward warping with nonparametric inpainting in our main paper.

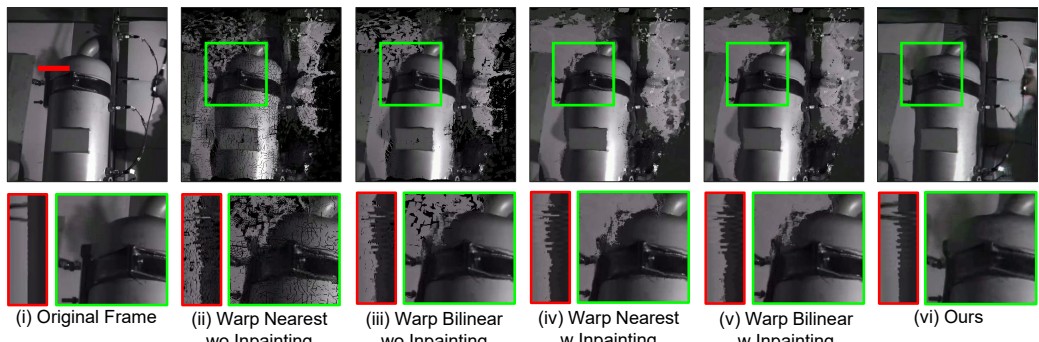

|  |  |  |  |  |  |
|---|---|---|---|---|---|
| (i) Original Frame | (ii) Warp Nearest wo Inpainting | (iii) Warp Bilinear wo Inpainting | (iv) Warp Nearest w Inpainting | (v) Warp Bilinear w Inpainting | (vi) Ours |

Figure A4: **Failure Cases of Forward Warp Baselines.** We show one frame from the *boiler* sequence, along with the magnified frame from warp nearest/bilinear without inpainting, warp nearest/bilinear with nonparametric inpainting. The closeups and $y$-$t$ slices obtained through forward warp methods with and without inpainting exhibit difficulties in managing object boundaries and the background. In contrast, our method successfully produces a satisfactory magnified frame under these challenging conditions.

Table A7: **Evaluation on Real-world Videos for Different Inpainting Methods for Forward Warp.** Similar to Table 1, we evaluated and compared among different inpainting methods for both warp nearest and warp bilinear, including no inpainting (labeled as "None"), nonparametric inpainting, and DeepFillv2 [70] inpainting.

| Method | Inpainting | PWC-Net | | | | | RAFT | | | | | GMFlow | | | | |
|---|---|---|---|---|---|---|---|---|---|---|---|---|---|---|---|---|
| | | $\alpha$=2 | $\alpha$=4 | $\alpha$=10 | $\alpha$=16 | $\alpha$=64 | $\alpha$=2 | $\alpha$=4 | $\alpha$=10 | $\alpha$=16 | $\alpha$=64 | $\alpha$=2 | $\alpha$=4 | $\alpha$=10 | $\alpha$=16 | $\alpha$=64 |
| | None | 0.92 | 1.86 | 5.43 | 9.12 | 42.23 | 0.50 | 1.18 | 3.87 | 7.13 | 39.82 | 0.91 | 1.70 | 5.29 | 9.05 | 50.38 |
| Warp Nearest | Nonparametric | **0.59** | 1.42 | **4.15** | **7.24** | **35.76** | 0.40 | 0.83 | 2.91 | 5.49 | **28.87** | 0.56 | **1.26** | **3.84** | **6.86** | 33.61 |
| | DeepFillv2 | 0.61 | **1.40** | 4.21 | 7.44 | 36.20 | **0.39** | 0.84 | 2.96 | **5.47** | 29.21 | **0.55** | 1.27 | 3.91 | 6.95 | 33.63 |
| | None | 0.68 | 1.49 | 4.32 | 7.44 | 36.93 | 0.39 | 0.91 | 3.07 | 5.79 | 30.60 | 0.66 | 1.62 | 4.26 | 7.40 | 35.71 |
| Warp Bilinear | Nonparametric | **0.57** | **1.40** | **4.12** | **7.28** | **35.78** | 0.35 | **0.78** | 2.97 | 5.52 | **28.70** | **0.53** | **1.24** | **3.79** | **6.88** | 33.72 |
| | DeepFillv2 | **0.57** | 1.43 | 4.14 | 7.34 | 35.93 | **0.34** | **0.78** | **2.96** | **5.50** | 28.95 | **0.53** | **1.24** | 3.92 | 7.04 | **33.45** |

Table A8: **Evaluation on Synthetic Videos for Different Inpainting Methods for Forward Warp.** Similar to Table 2, we evaluated and compared among different inpainting methods for both warp nearest and warp bilinear, including no inpainting (labeled as "None" in this Table), nonparametric inpainting, and DeepFillv2 [70].

| Method | Inpainting | Motion Error ↓ | | | | | | Magnification Error ↓ | | | | | | SSIM ↑ | | | | | |
|---|---|---|---|---|---|---|---|---|---|---|---|---|---|---|---|---|---|---|---|
| | | Subpixel Test | | | Noise Test | | | Subpixel Test | | | Noise Test | | | Subpixel Test | | | Noise Test | | |
| | | 0.04px | 0.2px | 1px | 0.01x | 1.0x | 100.0x | 0.04px | 0.2px | 1px | 0.01x | 1.0x | 100.0x | 0.04px | 0.2px | 1px | 0.01x | 1.0x | 100.0x |
| | None | 36.20 | 3.74 | 2.25 | 0.71 | **0.89** | 207.76 | 1238.97 | 28.68 | 3.19 | **0.16** | **0.23** | 1.95 | 0.19 | 0.44 | 0.75 | 0.87 | 0.71 | 0.03 |
| Warp Nearest | Nonparametric | **10.60** | 3.01 | **2.12** | **0.65** | 0.91 | 221.32 | 209.10 | 22.61 | **3.12** | **0.16** | **0.23** | 1.90 | **0.77** | **0.92** | **0.97** | **0.97** | **0.85** | **0.18** |
| | DeepFillv2 | 10.66 | **3.00** | 2.13 | 0.66 | 0.91 | 220.66 | **206.04** | **22.39** | 3.13 | 0.16 | 0.24 | 2.06 | 0.72 | 0.91 | **0.97** | **0.97** | **0.85** | 0.14 |
| | None | 13.09 | **3.02** | 2.13 | 0.64 | **0.89** | 201.72 | 281.45 | **22.56** | 3.12 | **0.15** | **0.23** | 1.88 | 0.55 | 0.89 | 0.94 | 0.94 | 0.86 | 0.04 |
| Warp Bilinear | Nonparametric | 10.61 | **3.02** | **2.12** | **0.61** | 0.90 | 226.37 | 209.35 | 22.61 | **3.10** | 0.15 | **0.23** | 1.68 | **0.78** | **0.93** | **0.97** | **0.98** | **0.90** | **0.19** |
| | DeepFillv2 | **10.56** | **3.02** | **2.12** | 0.62 | 0.90 | 225.30 | **207.24** | 22.64 | 3.11 | **0.15** | **0.23** | 2.32 | 0.77 | **0.93** | **0.97** | **0.98** | **0.90** | 0.09 |

Table A7 and Table A8 present a detailed comparison between the forward warp baselines without inpainting and with the two inpainting methods. In our evaluation of real-world videos, we observed a notable decrease in motion error when using inpainting methods in comparison to the warp baselines without inpainting, as expected. On the other hand, during noise tests, especially when using large noise factors, the warp baselines without inpainting exhibited better performance. Notably, we detected an increase in SSIM with noise factor increasing (see Table A6 for more details), for warp baselines with inpainting in noise tests. Our examination revealed that warp baselines produced distorted frames with significant noise factors and struggled to magnify motion when input motion was minimal. Consequently, the evaluation for forward warping baselines may not be able to reflect the actual performance in such cases. Therefore, given the slight advantage of nonparametric inpainting on real-world evaluations, we show the baselines with nonparametric inpainting in visualizations.

**Discussion.** The two forward warp methods, implemented with advanced techniques as the motion estimator and the inpainter, serve as powerful baselines for Lagrangian techniques, effectively establishing an upper performance limit for straightforward Lagrangian methods. The magnified frames produced through forward warping closely resemble the input frames, thus exhibiting favorable image quality due to their inherent implementation characteristics. Nevertheless, our method consistently outperforms or matches these baselines across various evaluation metrics. Upon qualitative assessment, our approach excels as well, particularly in complex scenarios involving occlusion and

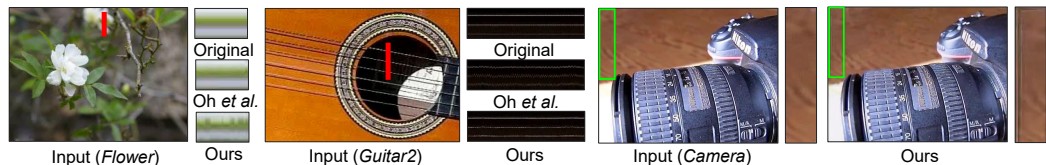

Figure A5: **Failure Cases.** We report several typical failure cases of thin structures or background artifacts. See Section A8 for details.

disocclusion or challenging flow patterns. We provide one challenging case in Figure A4, where large motion and occlusion were involved. While the results from forward warping methods with/without inpainting suffer from handling the object boundary and background, our method is still capable to provide a satisfactory magnified frame.

## A8    Failure Cases

While our method demonstrates favorable performance in most videos, there are cases where it falls short. One particular failure mode occurs when a video contains thin structures such as tree branches and guitar strings. In these cases it appears the flow network has incorrectly estimated the optical flow during training, causing our model to either "bleed" motion into the stationary background or fail to magnify the motion at all (see *flower* and *guitar* in Figure A5). The failure may be attributed to smoothness assumptions of flow estimators [50, 49, 23], and we expect results to improve as optical flow methods get more accurate. Another common failure is incorrectly magnifying background that has zero flow. This happens when small errors from the optical flow model are magnified, as shown in the *camera* sequence in Figure A5.

