# OpenReview forum: "Self-Supervised Motion Magnification by Backpropagating Through Optical Flow"
_NeurIPS.cc/2023/Conference — NeurIPS 2023 poster_

### Official Review · Reviewer_57E8 · 2023-07-04

**Soundness:** 3 good
**Presentation:** 1 poor
**Contribution:** 3 good
**Rating:** 6
**Confidence:** 5

**Summary:**

The paper proposed Lagrangian motion magnification using the pre-trained optical flow network. Magnification loss induces that the optical flow of the magnified frame matches with the optical flow of a given frame by $\alpha$ times; color loss regularizes the color consistency between a given and magnified frames. Test-time adaptation improves the quality of magnification on out-of-domain. Experiments show competitive performance compared to the prior arts in terms of SSIM and the proposed evaluation metrics.

**Strengths:**

1. Simple and effective algorithm to train motion magnification network using the off-the-shelf optical flow network.
2. Given the off-the-shelf optical flow network, this approach enables the training on large-scale unlabeled videos.
3. Targeted magnification and test-time adaptation might provide a better user experience.
4. The proposed method seems to be independent of the architecture of the neural network.

**Weaknesses:**

1. The term should be used carefully. I am not sure that the proposed method can be named "self-supervised" because the off-the-shelf optical flow network, which is used in experiments, is trained by supervised learning. If authors want to use the term "self-supervised", the self-supervised optical flow network should be used in the main experiments and the supervised one would be the strong baseline to be compared; It is not sufficient that the self-supervised optical flow network can be used in theory.
2. Evaluation metric is limited. To justify the underperformance in SSIM, the authors notice this phenomenon in the last sentence in Table 3, "DeepMag explicitly trains for SSIM". The proposed algorithm is also optimized by the proposed evaluation metric, Motion Error. For the same reason, I cannot be convinced about the quantitative results.
3. It is better to include the limitation to use the optical flow network. The optical flow network is inferior to estimate the subtle motion. It is related to the underperformance in the 0.04px subpixel test of Table 3. Thus, the limitation induced by the optical flow network should be investigated as an ablation study because this subtle motion is important in magnification.
4. I think that this proposed method is more general than supervised or self-supervised learning because this is determined by which optical flow network is used. How about using the synthetic data together?
5. Is the network architecture different from DeepMag? As for the control experiments, do the number of parameters affect the performance directly?
6. I think that DeepMag is sufficient as the strong baseline. However, I wonder why Warp Nearest and Bilinear are used, and the more advanced hand-crafted algorithms [A, B] are not used as baselines. It is because evaluation data might contain large motion?

[A] Phase-Based Video Motion Processing (SIGGRAPH 2013)

[B] Riesz Pyramids for Fast Phase-Based Video Magnification (CVPR 2014)

**Questions:**

My main concerns are the used term and fair comparison. See the weaknesses part.

=====

I update my rate from 4 to 6 because the authors will reflect the discussion below.

**Limitations:**

Limitations and broader impacts are described at the end of the main paper.

---

> ### Author Rebuttal · Authors · 2023-08-10
>
> Thank you for your constructive and comprehensive feedback. We would like to address your points below:
>
> **The Term “Self-Supervised”**
>
> We term our method “self-supervised” because it does not need ground truth magnified motions, as opposed to previous work DeepMag. It is common practice in the ML and vision communities to label methods self-supervised even if they use supervised components somewhere in the system, particularly optical flow. Below are papers presented at reputable conferences that follow this standard:
>
> - "Self-Supervised Learning of Motion Capture." H. Fish Tung, H. Tung, E. Yumer, K. Fragkiadaki. NeurIPS 2017.
>     - Self-supervised objective for predicting a 3D body mesh from a video, with a term that uses optical flow computed using FlowNet2.0.
> - "The Sounds of Motion." H. Zhao, C. Gan, W. Ma, A. Torralba. ICCV 2019.
>     - Self-supervised method for sound source separation with motion as auxiliary information, extracted from a pretrained PWCNet.
> - "Self-Supervised Learning via Conditional Motion Propagation." X. Zhan, X. Pan, Z. Liu, D. Lin, C. Loy. CVPR 2019.
>     - Self-supervised pretext task in which a network is required to predict optical flow from sparse motion signals, both computed from the supervised LiteFlowNet.
> - "Self-Supervised Learning of Audio-Visual Objects from Video." T. Afouras, A. Owens, J. Chung, A. Zisserman. ECCV 2020.
>     - Self-supervised loss to find "audio-visual objects," which utilizes tracks computed using PWC-Net.
> - "Self-Supervised Representation Learning from Flow Equivariance." Y. Xiong, M. Ren, W. Zeng, R. Urtasun. ICCV 2021.
>     - Learns representations by enforcing equivariance to a warping under optical flow, computed from RAFT.
>
> In addition, we train a version of our model using ARFlow, a fully unsupervised optical flow model, and present results in the attached PDF in Tables 2 and 3. Despite never fully finishing training, it performs well on metrics and beats DeepMag a significant amount of the time. We will include completed results in our manuscript.
>
> **Evaluation Metrics**
>
> We would be curious if the reviewer has an alternative metric in mind. Motion magnification is an incredibly hard task to measure quantitatively with no standard benchmarks, but with many possible metrics with different trade-offs. As such, we aim to present a holistic, fair, and well-rounded view of the performance of our models and baselines.
>
> To do this we use both SSIM, which favors DeepMag and is used in their evaluations, and we introduce the Motion Error metric, which favors our method. We mitigate the advantage our model by using *three* optical flow models to compute the Motion Error: RAFT which is used during our training, and PWC-Net and GMFlow which is never used by our method at all. Our method outperforms DeepMag on all instances.
>
> In addition, DeepMag has a significant advantage in the synthetic evaluations as it is being tested on in-domain data. This is because the process used to generate the synthetic train and test set are almost identical (both use PASCAL VOC objects on COCO backgrounds undergoing only translations). Despite this advantage we are still able to outperform DeepMag on the the Motion Error metrics and perform well on the SSIM metrics.
>
> **Very Small Flows**
>
> We agree that our method is closely tied to the performance of the optical flow method used during training and will add more discussion of this in the limitations section. On the other hand, because our method does not rely on a specific optical flow method it is free to take advantage of future progress in optical flow models.
>
> As for ablations on small motions, we point out that we already investigate small sub-pixel motions in depth in table A3 of the appendix where our method outperforms DeepMag on motion error. If there are specific ablations that you believe would be helpful please let us know.
>
> **Joint Supervised and Self-Supervised Training**
>
> Our method can certainly be combined with existing supervised approaches. However, our primary goal in this paper was to investigate our motion magnification objective. For this reason we opted to focus on the methods in isolation as combining them could make evaluation difficult. It would not be clear how to disentangle the benefits from different objectives.
>
> **Architectural Differences**
>
> Yes, the architectures are different. Because our method is not tied to a specific architecture we use the simple and ubiquitous UNet. DeepMag uses a bespoke architecture designed specifically for motion magnification, consisting of an encoder, a decoder, and a manipulator. The encoder encodes frames into a "texture representation" and a "shape representation" and the manipulator modifies the shape representation to achieve motion magnification.
>
> In addition, we show results in the attached PDF of a smaller model in Tables 2 and 3. This model has 1.09 million parameters compared to DeepMag’s 0.92 million. Due to time constraints we were only able to fully train the model, but despite this it performs similarly to our original model as compared to DeepMag. We will add completed results to the paper.
>
> **Forward Warp Baselines**
>
> The forward warp baselines we use are the simplest Lagrangian motion magnification methods. Therefore these methods serve as a point of reference, or a lower bound, for the reader. In addition, they give a sense of how well using *just* optical flow can perform, as opposed to using it in our proposed objective.
>
> **Phase Based Baselines**
>
> DeepMag shows superior performance against the phase-based methods so we believed it sufficient to compare against DeepMag. Moreover, evaluation of phase-based methods typically requires application of a temporal filter which makes it quite tricky to design a fair comparison. For an example of this please see the DeepMag paper.
>
> &nbsp;
>
> Thank you for your thorough review, and please let us know if we can further answer any questions you may have.

---

> > ### Comment · Reviewer_57E8 · 2023-08-14
> > **Response of Rebuttal by Authors**
> >
> > Thank the authors for responding to the comments.
> >
> > **The Term "Self-Supervised"**
> >
> > I am not sure and convinced about the authors' opinion. For meaningful discussion, I want to hear the author's opinion about the below example.
> >
> > Let's think about weakly-supervised semantic segmentation [A]. I think that this is similar setting of this paper.
> > - Train Model A on task A by supervised learning
> >   - This paper: Optical flow
> >   - Weakly-supervised semantic segmentation: Classification
> > - Apply Model A on ML algorithm B not using the ground truth of Task B
> >   - This paper: Motion Magnification
> >   - Weakly-supervised semantic segmentation: Segmentation
> >
> > In my opinion, "Weakly-Supervised Learning" is more appropriate to this paper because this work exploits more cheap ground truth of optical flow rather than motion magnification. Can authors provide their opinion about this? My big concern is the use of terminology in this work. If this is tackled, I will raise my score.
> >
> > And where can I find the result of the method trained by ARFlow in Tables 2 and 3? I would appreciate for the author to provide the line number.
> >
> > [A] Evaluation for Weakly Supervised Object Localization: Protocol, Metrics, and Datasets, TPAMI 2022
> >
> > **Evaluation Metrics**
> >
> > **(i)** I understand the difficulty of evaluating motion magnification, like the evaluation of a generative model. Since there exists no public benchmark of motion magnification, it might be difficult to request the construction of real-world dataset; move the object by x and alpha x exactly and make data pair.
> >
> > I think that "We do better on motion error, but lag slightly behind on SSIM, which DeepMag explicitly trains for" should be removed for the following reasons:
> >
> > (1) Question about the superior performance on motion error because the authors' method is trained for this,
> >
> > (2) Doubt of the validity of the evaluation metric from (1).
> >
> > I believe that it does not degrade authors' work.
> >
> > **(ii) Joint Supervised and Self-Supervised Training**
> >  This question is followed by the performance number of motion errors and SSIM. It is because joint training will improve both metrics together. I agree with authors' response.
> >
> > **Architectural Differences**
> >
> > I think that reporting the number of parameters is sufficient though the experiment is the direct way.
> >
> > **Additional paper**
> > I suggest this paper: "Video Motion Magnification to Improve the Accuracy of Vision-Based Vibration Measurements".
> > It is because this paper can support the need for motion magnification, and the learning-based method would be superior compared to other methods.

---

> > > ### Comment · Reviewer_57E8 · 2023-08-15
> > > **Response of Rebuttal by Authors**
> > >
> > > I checked Table 2 and Table 3 from the attached pdf.
> > > In my opinion, if the results are replaced by ARFlow, I respect the self-supervised method.
> > > In addition, the results in the main paper can be the strong baseline or upper bound.
> > >
> > > I believe that the number might not be important compared to DeepMag in this case, and the approach is cool.
> > > This method is more general, so the trained method, such as supervised, self-supervised, and weakly supervised does not degrade this idea. However, if the definition or terminology is arguable, I think that It should be discussed carefully.
> > >
> > > Can authors provide a discussion about this point?

---

> > > > ### Author Response · Authors · 2023-08-15
> > > >
> > > > Thank you for reading and responding to our comments!
> > > >
> > > > **The Term “Self-Supervised”**
> > > >
> > > > In light of the unsupervised ARFlow results would the following changes to the manuscript be satisfactory?
> > > >
> > > > - Include fully trained ARFlow results under the label of fully self-supervised
> > > > - Keep the RAFT results, but refer to them as “weakly supervised.” In addition indicate that they serve as a rough upper bound for a fully self-supervised method.
> > > > - Include a careful discussion of self-supervision and weak supervision in the context of our proposed method
> > > >
> > > > We apologize for any confusion and please let us know what you think.
> > > >
> > > > **Evaluation Metrics**
> > > >
> > > > We agree with your assessment that Motion Error is not necessarily the better metric and SSIM is not necessarily deficient. Both metrics have different trade-offs and are useful in different scenarios. We can remove the phrase you mention and include an in-depth discussion of the metrics that we use.
> > > >
> > > > **Architectural Differences**
> > > >
> > > > We will provide more details on the architectures used in our paper, including the number of parameters.
> > > >
> > > > **Additional References**
> > > >
> > > > Thank you for the paper reference! We will include this in our bibliography and discuss it in our related work section.
> > > >
> > > > &nbsp;
> > > >
> > > > Again, thank you for your comments. Please let us know your thoughts on the above proposed edits to the manuscript. We would be happy to discuss any other points you may have!

---

> > > > > ### Comment · Reviewer_57E8 · 2023-08-20
> > > > >
> > > > > I appreciate the authors' response. If the authors clear the confusion as the authors mentioned above, it would resolve my concerns, and I will update my rating.

---

> > > > > > ### Author Response · Authors · 2023-08-21
> > > > > >
> > > > > > Thank you so much for your feedback! We note that the NeurIPS 2023 submission policies do not allow us to update our manuscript during the rebuttal process, but we will certainly make the changes noted above and clear any confusion with respect to supervision in our manuscript. We thank you for your time and thoughtful feedback, and your willingness to update your rating.

---

### Official Review · Reviewer_hFGm · 2023-07-05

**Soundness:** 3 good
**Presentation:** 3 good
**Contribution:** 2 fair
**Rating:** 5
**Confidence:** 3

**Summary:**

This paper proposes a self-supervised model to solve the Lagrangian motion magnification problem without needing ground-truth labels. The network takes as input the two input frames and a magnification factor that ranges from 1 to 16, and outputs a generated frame that has magnified motion from the first frame. Off-the-shelf optical flow networks are used in loss computation for self-supervision. Test-time adaption has been explored to enhance the generation quality. Experiments show promising results.


**Strengths:**

1. Good writing; overall clear. The studied problem has received arguably less attention in the research community, but the related work section is detailed and well-structured, which especially helps the readers to catch up.
2. The method is very simple and easy to understand.
3. Experiments show promising results.
4. The authors promised to release full code upon acceptance.

**Weaknesses:**

1. The targeting application of this task is not clear. Why is this task important? Which data domains or scenarios do we want it to work? If our goal is just to detect small motions, we can develop optical flow estimation methods that work specifically for small motions. Even for existing state-of-the-art optical flow networks, detecting small motions is generally not a big issue, and it should not be hard to find a way to visualize small optical flow. Why do we need to generate a video? Maybe adding some application examples in the introduction and some results on related datasets will help the reader better understand the background and goal of this task.
2. There are still some confusions on the method. See questions below.
3. Some minor edits. See additional comments below.

**Questions:**

1. As you included the optical flow network in the whole computation graph with gradient computations, did you freeze its weights to make sure it does not shift away? If so, please state so explicitly in the paper.
2. Line 185-186: Why is the positional encoding conditioned on the factor $\alpha$? How do you do it? It does not make sense to me if no explanations are given.
3. Test-time adaptation: this trick adds in overhead on inference time. How efficient is it?
4. Did you tackle the occlusion issue? I believe this is an issue both for new frame generation and optical flow estimation. A simple warping using optical flow may be good enough if there are no occlusions at all.

Additional comments:
1. Line 26-28:  "The datasets and learning procedures that are used by these models are designed to be general-purpose, with a particular focus on ensuring that they apply to a variety of motions, objects, and scenes". Your cited methods are all supervised methods, which are usually trained on large synthetic datasets that could be totally different from real use cases. The generalization ability of these models is still an open question. I think a better idea is to weaken this statement.
2. Line 32-34: "And generate a new image pair whose predicted optical flow is $\alpha$ times as large as that of the input". Maybe add that the new image pair should share the same reference frame from the input, to avoid confusions.
3. Line 33-34: add $(\alpha \geq 1)$ to be more clear. Your method also makes sense even if $\alpha < 1$, so it is better to clarify your work cases.
4. Line 38: "it" -> "our method".
5. Line 61: need a citation for "Eulerian approaches".
6. Line 69: "it uses it" -> "we use it".
7. Line 115: It is better to make the symbol $x$ bold like $\mathbf x$ since it is a vector.
8. Eq 1: Maybe add that this equation assumes no occlusions.
9. Fig 6 caption: "subset" -> "subsets".
10. Repeated references: [21] and [22], [37] and [38].

**Limitations:**

Maybe need to add occlusions as a key limitation

---

> ### Author Rebuttal · Authors · 2023-08-10
>
> Thank you for your thorough and insightful review. We would like to address your questions as follows.
>
> **Importance of Motion Magnification**
>
> Motion magnification has applications in fields ranging from medical imaging to structural engineering to micro-expression analysis. Motion magnification enables us to amplify subtle movements thereby revealing dynamics that may have previously been hidden or hard to interpret. In the realm of medical imaging motion magnification may be used to magnify minuscule motions within tissues or organs to assist medical experts for diagnosis. For engineers motion magnification may serve as a powerful tool for detecting structural faults or weaknesses in buildings, bridges, or machinery. Motion magnification can also be applied to emphasize emotions, actions, or facial expressions that would otherwise be too small to detect.
>
> **Importance of Magnification Targeting**
>
> To magnify the motion of certain objects instead of motion in the entire frame, previous phase-based work used temporal filtering to select motion within specified frequency bands. This is a rather unintuitive interface to select objects for magnification and is used mainly because frequency filtering is particularly natural for phase-based methods. In addition, there may be multiple objects in a video moving at similar frequencies making it hard to isolate each object individually.
>
> We propose “targeted magnification” as a natural and intuitive way to allow users to magnify  objects within a segmented area. With recent advances in object segmentation it is much easier and straightforward for users to click and choose to magnify specific objects, giving precise control over what areas of the video to magnify.
>
> **Directly Using Optical Flow for Motion Detection**
>
> Visualization of optical flow is generally hard to decipher and motion magnification offers a much better alternative. Colorwheel visualizations where directions are mapped to arbitrary colors can be unintuitive to understand, especially when the motion changes quickly and many colors flash by rapidly. In addition, the magnitude of the motion is typically encoded by the saturation and is thus hard for humans to make fine differentiations between flow magnitudes.
>
> Overall, motion magnification is an elegant solution to the problem of detecting, visualizing, and understanding small motions in videos. We will include a discussion of these points in our paper. Please let us know if you have further questions.
>
> **Frozen Optical Flow Network**
>
> Yes, the optical flow network is frozen but is still backpropagated through to give a learning signal to the generation network. We will make this explicit in the paper.
>
> **Alpha Encoding**
>
> In order to pass the desired magnification factor, $\alpha$, to our network we use a "positional embedding" scheme to encode the scalar $\alpha$ into a vector. The details of this implementation can be found in Section A2 of the appendix as well as in the code from the supplementary materials. In particular, we use a standard sinusoidal embedding scheme where alpha is passed through a series of sinusoids with exponentially increasing wavelength and we stack the result into a vector. We apologize for any confusion and will edit the text in our paper so that this is clearer.
>
> **Test Time Adaptation Efficiency**
>
> The efficiency of the test time adaptation experiments presented in the paper were not specifically optimized, nor was their performance measured. We rerun the experiments corresponding to the three examples from Figure 2 in the paper ("baby," "cats," and "pole") to provide more careful measurements of test time adaptation efficiency. Qualitative visualizations of improvement and quantitative measures of efficiency are in the attached PDF in Figure 1 and Table 1. We find that even with a thousand gradient steps and tens of minutes, improvement can be seen on out of distribution data. We give a precise explanation of the test time adaptation procedure below, which will be added to the manuscript.
>
> Given a video for test time adaptation we construct a dataset using every frame, with minor rotation and color augmentation. We then train for N epochs, where an epoch consists of all the frames in the video. The learning rate is set to be 1e-4 (a third the learning rate during train time) and the batch size is 8. Note that the training times vary based on the number of frames and the size of the video.
>
> **Occlusion and Disocclusion**
>
> This is an excellent observation. As can be seen in the supplemental video our model produces qualitatively good results even in the presence of occlusions and disocclusions. This is because optical flow models are already designed to handle occlusions and disocclusions well, so our magnification loss also incentivizes correct predictions in these scenarios.
>
> The other component in our loss, the color loss, can be affected by occlusions and disocclusions. To mitigate this, in early experiments we tried masking occlusions with our loss, where we check for occlusions by computing flow both forwards and backwards and performing a cycle consistency check. We apply this occlusion mask to the color loss such that pixels that are occluded or disoccluded do not contribute to the loss. We found that results did not change significantly with occlusion masking, possibly because the magnification loss already handled occlusions and disocclusions adequately, and we therefore omitted this from our final method.
>
> Overall, driven by our “magnification loss” we are able to handle occlusions and disocclusions adeptly as evidenced by our qualitative results. We will revise our manuscript to include a discussion of this point.
>
> **Typos and Suggestions**
>
> Thank you for findings these. We will revise our paper to include your suggestions and fix typos.
>
> &nbsp;
>
> Thank you for your thoughtful review, and please let us know if you have further questions or need further clarification.

---

> > ### Comment · Reviewer_hFGm · 2023-08-10
> >
> > Thanks for the response! I still have questions on the following topics.
> >
> > **Importance of Motion Magnification**
> > If the goal of the task is to detect and highlight small motions in different applications, we could just visualize the magnitude of the optical flow field. Ideally, most part of the visualization will be totally black due to zero motion, and only the small motion part is highlighted. This should be obvious enough if the goal is just to detect small motions, so why do we need to generate a new video with magnified motion? For medical workers or engineers, this type of visualization should be easier to read than original color images.
> > In addition, generating a new video with magnified motion will most likely create new occlusions, so a part of the information in the original input may be lost. For example, what if the magnified motion covers a part of the background that also contains another small motion that needs to be detected? In comparison, a simple visualization of per-pixel motion such as optical flow should be able to highlight every small motion in the image.
> > I suggest that the authors could use some examples from targeting applications as demo, instead of the current "baby" and "cats" examples (looking like naive toy cases). That will help better explain the background applications of the task.
> >
> > **Occlusion and Dis-occluson**
> > Based on my experiences in optical flow estimation, occlusion is still a very challenging issue and also a major source of error for most of the latest optical flow models including RAFT. The correspondences of occluded pixels are not perceivable in the second frame, so the model can only "guess" the correct flow based on smoothness and other cues. I think the main reason that occlusions do not hurt too much in your case is that the occlusion regions are also very small if the motions are generally small, so masking the occlusion region does not make big differences. Maybe you could visualize your occlusion masks and argue in this direction. Anyway, stating an optical flow model can handle occlusions well could be a bit dangerous.
> > Still, the paper should acknowledge the issues on occlusions even if you could argue it does not hurt your specific task too much. Many of the equations and losses do not work at occlusion regions, so it will be confusing if you do not mention occlusions there at all.

---

> > > ### Author Response · Authors · 2023-08-13
> > >
> > > **Importance of Motion Magnification**
> > >
> > > We agree with you that if our goal were to detect or localize small motions we could just plot an optical flow field. The goal of motion magnification is instead to visualize these small motions. For very simple motions, this may not be necessary and your proposal would work well, but often people need to understand complex motions that can’t be visualized well from optical flow alone. For example, in Figure 2c of our paper we motion magnify the “pole” clip, originally from [1]. Looking at the patch visualization it is clear that the column is vibrating, but more precisely motion magnification shows that it is vibrating at *two different modes*. One lower frequency mode at a higher amplitude, and one higher frequency mode at a lower amplitude. The superposition of the two modes produces the distinct “squiggly sine wave” patch visualization. Understanding this from just optical flow visualization alone would be quite difficult, but magnifying motions makes this abundantly clear. Other examples of motion magnification visualization can be found in the cited PNAS article [2], where the authors use and validate the technique for a number of scientific applications. This includes visualization of the modal shapes of a pipe (Fig. S3) and a lift bridge (Fig. 3), vibrations in ear tissue (Fig. 2), and deformations in a metamaterial under forcing (Fig. 4).  In addition, we point out that at this year’s SIGGRAPH conference the original Eulerian motion magnification paper [3] was awarded the “Test-of-Time Award,” highlighting the impact of this visualization technique. Finally, with respect to your suggestion of replacing the “baby” sequence in our paper, we note that it is a "classic" example used in prior work [3,4,5,6].
> > >
> > > We will edit our draft to better explain the applications of motion magnification and include more practical applications of the technique, and we thank you for your suggestions.
> > >
> > > **Occlusion and Disocclusion**
> > >
> > > We also agree with you that occlusions and disocclusions are a challenge in the context of motion magnification methods. Our discussion above was not meant to imply that we’ve solved the problem of occlusions, but rather to point out our method does not ignore occlusions entirely and is able to reason about them through the optical flow model. As you correctly point out, these models are not perfect at predicting flow in occluded areas and therefore large occlusions may be challenging. However, because our method is agnostic to the form of tracking used we believe it can benefit from future progress in optical flow estimation. Additionally, empirical results demonstrate that motion is amplified well despite these concerns. This is likely due to your point that motions we consider are small. Overall, we thank you for your helpful thoughts on this problem and your insightful recommendations. We will revise our draft to include a thorough discussion of these points.
> > >
> > > &nbsp;
> > >
> > > Again, thank you for your suggestions and please let us know if you have any remaining questions or would like to discuss a point further!
> > >
> > > &nbsp;
> > >
> > > [1] “Structural Modal Identification through High Speed Camera Video: Motion Magnification.” Justin G. Chen, Neal Wadhwa, Young-Jin Cha, Frédo Durand, William T. Freeman, Oral Buyukozturk. *Proceedings of the 32nd International Modal Analysis Conference (2014).* http://people.csail.mit.edu/mrub/vidmag/papers/Chen_Imac_2014.pdf
> > >
> > > [2] “Motion microscopy for visualizing and quantifying small motions.” Neal Wadhwa, Justin G. Chen, Jonathan B. Sellon, Donglai Wei, Michael Rubinstein, Roozbeh Ghaffari, Dennis M. Freeman, Oral Büyüköztürk, Pai Wang, Sijie Sun, Sung Hoon Kang, Katia Bertoldi, Frédo Durand, and William T. Freeman. *Proc. Natl. Acad. Sci., 114 (44) (2017), pp. 11639-11644.* https://www.pnas.org/doi/full/10.1073/pnas.1703715114
> > >
> > > [3] “Eulerian Video Magnification for Revealing Subtle Changes in the World.” Hao-Yu Wu, Michael Rubinstein, Eugene Shih, John Guttag, Frédo Durand, William T. Freeman. *ACM Transactions on Graphics, Volume 31, Number 4 (Proc. SIGGRAPH), 2012*.
> > >
> > > [4] “Phase-based Video Motion Processing.” Neal Wadhwa, Michael Rubinstein, Frédo Durand, William T. Freeman. *ACM Transactions on Graphics, Volume 32, Number 4 (Proc. SIGGRAPH), 2013.*
> > >
> > > [5] “Riesz Pyramids for Fast Phase-Based Video Magnification.” Neal Wadhwa, Michael Rubinstein, Frédo Durand, William T. Freeman. *IEEE International Conference on Computational Photography (ICCP), 2014.*
> > >
> > > [6] “Learning-based Video Motion Magnification.” Tae-Hyun Oh*, Ronnachai Jaroensri*, Changil Kim, Mohamed Elgharib, Frédo Durand, William T. Freeman, Wojciech Matusik. *European Conference on Computer Vision (ECCV), 2018.*

---

> > > > ### Comment · Reviewer_hFGm · 2023-08-13
> > > >
> > > > Thanks for your detailed response! I now have a better understanding about the background of this task. I think this paper could be more accessible for more specialized conferences such as SIGGRAPH or CVPR. For NeurIPS audiences, it is better to include those technical backgrounds so that the readers could understand the goals and expectations of this task.

---

> > > > > ### Author Response · Authors · 2023-08-15
> > > > >
> > > > > Thank you so much for your reply and helpful suggestions to help us refine the paper. We will include relevant technical background in our manuscript and tailor it to the NeurIPS audience. If you have any additional suggestions to improve our paper please don't hesitate to share.

---

### Official Review · Reviewer_uXCg · 2023-07-06

**Soundness:** 3 good
**Presentation:** 3 good
**Contribution:** 3 good
**Rating:** 7
**Confidence:** 4

**Summary:**

The paper introduces an optical-flow-based lagrangian motion magnification method, learned through self-supervised learning. The architecture is very simple -- just a U-Net that inputs two temporally consecutive frames and outputs a motion-magnified image. To train the U-Net, the method uses an off-the-shelf optical flow method, estimates the motion between two frames, and considers the motion as real motion of the scene. Then it penalizes the difference between the estimated magnified motion (constant * estimated motion) and the motion between the reference image and motion-magnified image (i.e. output image). The method demonstrates both good quantitative and qualitative results.

**Strengths:**

+ Comprehensive related work

  The paper provides a comprehensive literature survey, which helps understand previous related work and where this paper positions among them.

+ Implementation details

  Sec. 4.1, Sec. 4.2, and supplementary material provide sufficient implementation details so that it's easy to understand the choices of hyper-parameters, training configuration, dataset curations, and training details. Parts of source codes are also included in the supplementary material, which all help reproduce the proposed methods.

+ In-depth evaluation

  Fig. 5, Fig. 6, Table 2, and Table 3 provide in-depth evaluation of the proposed method and related methods on real-world and synthetic videos. Given that there is no public benchmark and the difficulty of evaluation on this topic, the paper tries its best on providing sufficient evaluation.

**Weaknesses:**

- How to handle occlusion and disocclusion?

  It seems already stated in the limitation section, but I wonder if the method doesn't explicitly handle occlusion and disocclusion. If it doesn't, then can it be a problem? Does U-Net learn to handle them to some extent? When watching the supplementary video, the model doesn't seem to output hallucinated appearance around the disoccluded region, which seems good.

- Worse SSIM in Table 3

  In Table 3, compared to DeepMag, all metrics are better except for SSIM. I wonder why it's the case. What makes the DeepMag's SSIM better than the proposed method?

- Moving background?

  In the supplementary video (1m14s and 2m12s), I am wondering why the background has motion and it's moving? Is it due to that the optical flow method hallucinates motion in the background and it's used during the training? Can this problem be resolved without using the target segmentation mask?
  By the way, this is another question: what if $L_{mag}$ in Eq. (3) is applied to the target segmentation objects only and penalizes the background motion to be zero? Can it produce better results and prevent the background from moving?


There are some unresolved concerns but the strength outweighs the weaknesses for now. I would like to give Borderline Accept for now, but the rating could change after the discussion phase.

---
All concerns are resolved. Thus I am updating my rating to 7. Accept.

**Questions:**

- Usage of the discriminator loss? I wonder if using an adversarial loss from GAN would help more realistic appearance where flow is not accurate, occlusion happens, or artifacts occur.
- I wonder if the collected data or at least the information on train/test splits will be available if the paper is accepted.

**Limitations:**

- Probably another limitation would be that the method's success depends on the off-the-shelf optical flow and segmentation methods.

---

> ### Author Rebuttal · Authors · 2023-08-10
>
> We thank you for your positive and comprehensive assessment of our paper. Below, we address your questions and concerns:
>
> **Occlusions and Disocclusions**
>
> This is an excellent observation. As noted our model produces qualitatively good results even in the presence of occlusions and disocclusions. This is because optical flow models are already designed to handle occlusions and disocclusions well, so our magnification loss also incentivizes correct predictions in these scenarios.
>
> The other component in our loss, the color loss, can be affected by occlusions and disocclusions. To mitigate this, in early experiments we tried masking occlusions with our loss, where we check for occlusions by computing flow both forwards and backwards and performing a cycle consistency check. We apply this occlusion mask to the color loss such that pixels that are occluded or disoccluded do not contribute to the loss. We found that results did not change significantly with occlusion masking, possibly because the magnification loss already handled occlusions and disocclusions adequately, and we therefore omitted this from our final method.
>
> Overall, driven by our “magnification loss” we are able to handle occlusions and disocclusions adeptly as evidenced by our qualitative results. We will revise our manuscript to include a discussion of this point.
>
> **SSIM**
>
> There are two main advantages that DeepMag has when evaluated on SSIM. Firstly, DeepMag is trained with a loss that is relatively similar to SSIM. As a result it tends to score higher on this metric. Secondly, the SSIM metric is computed on the synthetic test set from DeepMag. This test set is generated in a procedure that is very similar to how the train set for DeepMag was generated, and is therefore quite in-domain for the DeepMag model. On the other hand, our model was trained on real videos and the test set is therefore significantly out-of-domain both in terms of content (derived from COCO and PASCAL) and motion (only translations). We note that despite this domain gap we are able to outperform DeepMag on the Motion Error metrics on the synthetic test sets and achieve good SSIM results.
>
> **Moving Background in Supplemental Video**
>
> We believe that you are correct that the shimmering background comes from optical flow errors. The background of the cat video, which is extremely out-of-focus and features bokeh artifacts, is out-of-distribution and therefore particularly hard for the optical flow model. Our model is closely tied to the performance of the underlying optical flow model, and we will add a discussion of this in our limitations section.
>
> **Adaptive Magnification Loss**
>
> Your suggestion of only applying the magnification loss to segmented objects should work in removing background motion, and would certainly be an interesting extension of our loss for future work. It would however also add complexity to the training process as it would be necessary to generate segmentations and make foreground-background predictions. In addition it could remove motion that the user might want to keep. Therefore for this paper we opt to keep our method simple and just train to magnify all motion, and then allow the user to choose at inference time which areas of the video to magnify through our targeting procedure.
>
> **GAN Loss**
>
> A GAN loss could certainly help the performance of the method. However, we wanted to focus this work specifically on the performance of the self-supervised loss that we introduce. To this end we opted to omit a GAN loss as it would make it hard to disentangle the roles of the adversarial loss and the motion magnification loss. Even without a GAN loss we are able to produce magnified sequences that are competitive with or outperform existing supervised methods, demonstrating the effectiveness of our proposed method.
>
> **Dataset Information**
>
> We are more than happy to release information about the dataset upon acceptance. In fact, enough data to perfectly reconstruct the dataset should already be in the paper and the supplemental material. Table 1 contains the temporal strides used to sample frame pairs from the 5 constituent datasets, in addition to the number of frame pairs before and after motion filtering. Appendix A3 contains additional information about how the constituent datasets were sampled, details on how the motion filtering was performed, and an overview of how the test set was constructed. We plan on releasing our datasets and all of our code upon acceptance, including the code used to sample, filter, and compile our train and test datasets.
>
> **Limitations of Off-the-Shelf Models**
>
> We agree with the observation that our method is fundamentally limited by the quality of off-the-shelf models, and will add this to our limitations section. We however also like to caveat that this dependence on off-the-shelf models means that our method can also benefit from improvements in optical flow models.
>
> &nbsp;
>
> Thank you for your insightful review, and please let us know if you have further questions or need further clarification.

---

> > ### Comment · Reviewer_uXCg · 2023-08-15
> >
> > Thanks for the detailed responses. It resolves my concerns. I will also read other reviews and update my rating accordingly after the discussion period!

---

> > > ### Author Response · Authors · 2023-08-18
> > >
> > > Thank you for your reply! Please do not hesitate to contact us if you have more questions.

---

### Official Review · Reviewer_nhiY · 2023-07-06

**Soundness:** 3 good
**Presentation:** 4 excellent
**Contribution:** 3 good
**Rating:** 6
**Confidence:** 4

**Summary:**

This paper uses the classical method of Lagrangian to self-supervise the task of motion magnification. Thanks to the proposed self-supervision-based technique, the proposed method can also be adapted during the test time. As shown in Figure.1, the proposed method is simple, where the optical flow vectors of videos before and after magnification are compared.  The optical flow of the motion amplified video is compared to the scaled (by the amplification factor) optical flow of the original video, to derive the magnification loss. To make the output video color consistent, color loss is also used. Videos are provided in the supplementary material for qualitative analysis.

**Strengths:**

1.	The method presented in this paper is simple, straightforward, and meaningful.
2.	The experimental evaluations validate the proposed method. Supplementary videos are helpful.
3.	The source code is also provided in the supplementary, which further highlights the simplicity.
4.	Limitations of the method are well discussed, and failure cases are shown.
5.	The paper is well-written and easy to follow.


**Weaknesses:**

1.	The proposed method largely depends on the pre-trained optical flow network.
2.	Given the nature of the addressed problem, its evaluation is known to be difficult. This is reflected in the experiments.
3.	The experiments are conducted in a relatively small amount of video frames, and the paper discusses “out of the distribution” and “test time adaptation”. It would be interesting to see how the method generalizes when training on a large number of videos, before proceeding to discuss the rest.


**Questions:**

How does the proposed method behave with low-quality optical flow? Does the method improve its performance in difficult videos, when trained on a larger amount of data? I can imagine that when trained on a large collection of videos, the method may generalize and hence also perform well in some difficult cases.

**Limitations:**

The authors adequately addressed the limitations.

---

> ### Author Rebuttal · Authors · 2023-08-10
>
> Thank you for your positive and well thought-out comments. We would like to address the questions that you have:
>
> **Pre-Trained Optical Flow**
>
> We agree with the assessment that our model’s performance is closely tied to the performance of the underlying flow model, and will add a discussion of this in our limitations section. But we also note that because our method is agnostic to the form of tracking used (as long as it is differentiable), it is also able to take advantage of future improvements in optical flow models.
>
> **Low Quality Optical Flow**
>
> Our goal was to produce the best self-supervised motion magnification method possible. As such we use one of the best off-the-shelf optical flow algorithms, RAFT, and did not conduct any experiments analyzing the consequences of using worse flow. We would guess that with lower quality flow the motion magnification capabilities of models trained with our loss will worsen. With poor enough flow, training may not even converge. Conversely, with better flow we expect our method to correspondingly get better.
>
> **Larger Amount of Data**
>
> We agree with you and believe that more data should help the performance and generalization of models trained with our loss. For example, in our paper we train on a decently sized dataset of frame pairs. However, the distribution of this data is not broad enough to cover all of our test videos and we therefore see that test time adaptation can further improve the quality of our magnification results. With diverse enough data we believe that test time adaptation would become less necessary, or provide a smaller benefit, as our model generalizes better.
>
> &nbsp;
>
> Thank you for your thoughtful questions, and please let us know if you have further questions or need further clarification.

---

> > ### Comment · Reviewer_nhiY · 2023-08-21
> > **Discussion follow-up**
> >
> > Thank you for your reply. I suggest to include the experiments with different quality of optical flow.
> > I think, these are simple experiments to conduct, therefore should be addressable in the camera-ready.
> > Hence, I keep my review towards accepting this paper.

---

> > > ### Author Response · Authors · 2023-08-21
> > >
> > > Thank you for your reply and suggestion. We will add these experiments. Please let us know if you have any other advice.

---

### Official Review · Reviewer_XnFN · 2023-07-06

**Soundness:** 4 excellent
**Presentation:** 4 excellent
**Contribution:** 3 good
**Rating:** 7
**Confidence:** 4

**Summary:**

This paper focuses on learning a pair-wise motion magnification model in a self-supervised manner. The authors employ recent optical flow models to estimate the flow fields between the original and the motion magnified image pairs. The UNet concatenates a sinusoidal encoded magnification factor with the original images and generates the magnified image.

The learning process of the UNet is facilitated by a loss function that enforces consistency between the original and magnified flow fields, as well as the consistency between the backward warped images. To demonstrate the effectiveness of the proposed method, the authors curate a large-scale real-world training set. They conduct evaluations both quantitatively on a synthetic dataset and qualitatively on real-world data. The results demonstrate superior performance over previous supervised methods learned on synthetic data.

**Strengths:**

- The paper is well organized and written. Sec.1 introduces the problem effectively and motivates the design of the method. Sec.2 provides a brief but comprehensive review of the previous methods. In addition to the overall organization, sufficient details (including the code) are given for a better understanding of the method, such as the footnote on page 5.

- The method itself is simple and effective:

  - A simple UNet is a compact solution that avoids complicated operations, e.g., explicit optical flow estimation and inpainting.

  - The magnification factor is concatenated with the input image pair after sinusoidal encoding, which enables regionally varying magnification. This is difficult for a single magnification factor as in [33].

  - The self-supervised learning losses enable online adaptation to a specific sequence for better quality.

- The evaluation is comprehensive and achieves significant improvement over previous methods.

- The curated large-scale real-world dataset will encourage further research in this topic.

**Weaknesses:**

I do not see significant weakness of the paper since it is simple and effective. The only missing piece I come up with is that since the model itself is simple, the author could make some deeper analysis of a learned UNet to understand the underlying mechanism of the model.


**Questions:**

- In lines 83-84, the authors claim that the method's capabilities are more similar to those of Lagrangian methods. However, the internal workings of the model remain unknown. It would be interesting if the authors could conduct some analysis, such as investigating whether some UNet layers implicitly contain motion cues.

- The warping loss compares warped images. What would happen if we additionally, or only, compare two warped images with I_0?

- In the supplementary video, is the magnification factor applied only to the ground in the jumping sequence?

**Limitations:**

The limitations have been addressed adequately in the paper.

---

> ### Author Rebuttal · Authors · 2023-08-10
>
> Thank you so much for your thoughtful and positive review. We appreciate that you believe our method to be simple yet effective, and are glad that you found our paper well-organized and well-written. Below, we address some of your questions and suggestions.
>
> **Analysis of UNet**
>
> We agree with you that a deeper analysis of the UNet could bring about better insight to how the model encodes motion information. Given that our model is able to successfully magnify motion there must be an understanding of this in the network. However, analyzing and interpreting internal activations of neural networks is still an open research problem. We believe our paper should focus primarily on our motion magnification objective, but are hopeful that our work could be a useful case study for future feature visualization or mechanistic interpretability research.
>
> **Lagrangian Label**
>
> We term our method Lagrangian because our loss explicitly tracks pixels through a video and tries to magnify those tracks, and not because of how the neural network behaves. This is as opposed to Eulerian approaches which aim to magnify motion by representing and manipulating motion as spatially fixed variables over time. These terms derive from Lagrangian and Eulerian frames of reference in the field of fluids.
>
> **Reformulation of the Loss**
>
> Given a generated image, which we hope to be motion magnified, our color loss warps this generated image such that it matches the target reference image $I_0$. We then take a photometric loss between the warped generated image and the reference image, with the goal of encouraging the network to generate images that have similar color to the reference image. To be consistent with prior work like DeepMag, our network only predicts a single magnified frame, but it should be possible to predict multiple frames. In this case we would compare two or more warped frames to the reference image. We do agree though that there is much room for extensions to our loss function for future work. Please let us know if this answered your question or if you would like anything else clarified.
>
> **Supplementary Video**
>
> The "jumping" clip appears twice in the supplementary video. In the first instance the magnification is applied to the entire clip, not just the ground. In the second instance we target the model to magnify the ground but not the legs. We remove magnification of the legs in the "jumping" clip because the motion of the legs is too large to be magnified reasonably at a magnification factor strong enough to see the ground shake.
>
> &nbsp;
>
> Again, thank you so much for your positive review, as well as your questions and feedback. We look forward to answering any more questions you may have.

---

> > ### Comment · Reviewer_XnFN · 2023-08-21
> >
> > Dear Authors:
> >   Thank you for the response and all my concerns have been addressed.

---

> > > ### Author Response · Authors · 2023-08-21
> > >
> > > Thank you for your reply! Please do not hesitate to contact us if you have more questions.

---

### Author Rebuttal · Authors · 2023-08-10

We thank all the reviewers for their insightful, thorough, and constructive reviews which have helped to improve our manuscript greatly. Please find individual responses to your questions and comments below. We look forward to the discussion period.

---

### Decision · Program_Chairs · 2023-09-21

**Decision:**

Accept (poster)

**Comment:**

While the initial reviews were mixed, this submission eventually received unanimously positive reviews from the reviewers after the discussion period. The AC agrees and recommends acceptance. The authors are encouraged to incorporate the feedback in the camera-ready version.